

# Integrated bagging-RF learning model for diabetes diagnosis in middle-aged and elderly population

Yuanwu Shi[*] and Jiuye Sun[*]

College of Art and Design, Wuhan Textile University, Wuhan, Hubei, China
[*] These authors contributed equally to this work.

## ABSTRACT

As the population ages, the increase in the number of middle-aged and older adults with diabetes poses new challenges to the allocation of resources in the healthcare system. Developing accurate diabetes prediction models is a critical public health strategy to improve the efficient use of healthcare resources and ensure timely and effective treatment. In order to improve the identification of diabetes in middle-aged and older patients, a Bagging-RF model is proposed. In the study, two diabetes datasets on Kaggle were first preprocessed, including unique heat coding, outlier removal, and age screening, after which the data were categorized into three age groups, 50–60, 60–70, and 70–80, and balanced using the SMOTE technique. Then, the machine learning classifiers were trained using the Bagging-RF integrated model with eight other machine learning classifiers. Finally, the model's performance was evaluated by accuracy, $F1$ score, and other metrics. The results showed that the Bagging-RF model outperformed the other eight machine learning classifiers, exhibiting 97.35%, 95.55%, 95.14% accuracy and 97.35%, 97.35%, 95.14% $F1$ Score at the Diabetes Prediction Dataset for diabetes prediction for the three age groups of 50–60, 60–70, and 70–80; and 97.03%, 94.90%, 93.70% accuracy and 97.03%, 94.90%, 93.70% $F1$ Score at the Diabetes Prediction Dataset. 95.55%, 95.13% $F1$ Score; and 97.03%, 94.90%, 93.70% accuracy; and 97.03%, 94.89%, 93.70% $F1$ Score at Diabetes Prediction Dataset. In addition, while other integrated learning models, such as ET, RF, Adaboost, and XGB, fail to outperform Bagging-RF, they also show excellent performance.

# INTRODUCTION

## Motivation

Diabetes mellitus, as a chronic metabolic disease, has become one of the most significant challenges to global public health. With the aging of the population, the prevalence of diabetes in the middle-aged and elderly population, a group with a high prevalence of diabetes, is significantly higher than that of other age groups (*WHO, 2024a*; *WHO, 2024b*; *WHO, 2024c*). Type 1 diabetes is caused by insulin destruction, while type 2 diabetes is caused by insufficient insulin secretion, leading to elevated blood glucose. Type 1 diabetes is an autoimmune disease that usually develops in childhood or adolescence and results

Corresponding authors
Yuanwu Shi, 645688909@qq.com
Jiuye Sun, hjhjn77@qq.com

in insufficient insulin production. Type 2 diabetes, on the other hand, is related to age, weight, and lifestyle and is the most common type of diabetes among older people, mainly due to the body's weakened response to insulin. Both types of diabetes, if left uncontrolled, can lead to serious complications such as heart disease, stroke, kidney disease, and vision loss. Chronic poor blood sugar control may also increase the risk of amputation and nerve damage.

Middle-aged and older adults are more likely to have a high prevalence of diabetes due to physiological decline and a poor diet. With the aging of the population, the growing group of middle-aged and elderly diabetics poses new challenges to the allocation of resources in the healthcare system. The development of accurate diabetes prediction models not only improves the efficiency of healthcare resource use and ensures that resources are prioritized and allocated to the neediest patients but also provides timely and effective care and treatment for middle-aged and older adults, which is a critical strategy in public health to address the challenges of aging.

Therefore, it is crucial to predict and manage diabetes in middle-aged and older adults. In recent years, machine learning technology has been increasingly used in healthcare, and its powerful data processing capabilities and pattern recognition functions have provided new perspectives and tools for disease prediction.

Machine learning has achieved remarkable results in predicting diabetes and middle-aged and elderly diseases, and current research mainly uses integrated models, neural networks, and deep learning to improve prediction accuracy. However, current research focuses on predicting the whole population and neglects the particular diabetes prediction of high-risk groups such as middle-aged and older adults. Therefore, future research should pay more attention to high-risk groups and try to use more kinds of classifiers for prediction.

## Objective

This study focuses on the high prevalence of diabetes, *i.e.,* the middle-aged and elderly population aged 50–80, and proposes the Bagging-RF integrated learning model. The Diabetes Simple Diagnosis and Diabetes Prediction Dataset datasets from the Kaggle website were selected for the experiment (*Kaggle, 2023*; *Kaggle, 2024*).

The data were first preprocessed, including uniquely hot coding, outlier removal, and age screening, to categorize the dataset into three age groups according to age: 50–60 years, 60–70 years, and 70–80 years. Due to the uneven distribution of the preprocessed data, the dataset was balanced by a few synthetic class samples using the SMOTE technique. Then, it was trained using the Bagging-RF integrated model and RF, GNB, LR, KNN, DT, ET, Adaboost, XGB, and eight machine-learning classifiers. The performance of each classifier is compared by metrics such as accuracy, precision, recall, specificity, AUC, and $F1$ score.

The results showed that the Bagging-RF integrated model demonstrated excellent performance in predicting diabetes in three age groups: 50–60, 60–70, and 70–80 years old. The study can effectively help community physicians identify high-risk individuals so as to take precautionary measures and make early diagnoses in advance, reducing the incidence of diabetes mellitus and the occurrence of complications.

## Strengths and weaknesses of this study

The main contributions of this study are as follows:

1. An integrated Bagging-DT-RF model is proposed, which shows excellent performance in predicting diabetes in all three age groups: 50–60, 60–70, and 70–80 years.
2. Experiments were conducted on the Diabetes Simple Diagnosis and Diabetes Prediction Dataset using KNN, LR, GNB, RF, ET, DT, Adaboost, and XGB machine learning classifiers, where the integrated models such as RF, ET, Adaboost, and XGB performed well.
3. Key metrics such as accuracy, precision, recall, specificity, $F1$ score, confusion matrix, ROC curve, and AUC are comprehensively applied to evaluate the performance of each machine learning classifier in the diabetes classification task.
4. The effect of the SMOTE data balancing technique in diabetes prediction models was analyzed to assess its impact on model performance.

The shortcomings of this study are as follows:

1. The proposed model's performance still needs improvement. Although the model performs well in predicting diabetes in 50–60-year-olds, there is still room for improvement in the accuracy of diabetes prediction in 60–70-year-olds and 70–80-year-olds.
2. The proposed model could be more interpretable, making it difficult for healthcare professionals and patients to understand its decision logic and prediction results.
3. The SMOTE technique may introduce noise that negatively affects the model's performance.

## Synopsis of the remaining chapters

The rest of the article is as follows: the literature review reviews the research of previous authors in machine learning diabetes prediction. The research methodology section describes the process and methodology used in this study. The experimental results section discusses the experimental results. The results discussion section discusses and analyses the experimental results. The summary section summarises the paper.

## LITERATURE REVIEW

The application of machine learning in diabetes and disease prediction in middle-aged and older adults has received much attention and has achieved significant results. The research mainly uses integrated models, neural networks, deep learning, data balancing techniques, and feature engineering to improve prediction accuracy, which is valuable for clinical practice.

The integrated learning model again shows excellent performance in diabetes prediction. *Tripathi et al. (2023)* trained on the PIMA dataset using the integrated model along with soft polling for diabetes prediction. When comparing five machine learning models, such as LR, KNN, SVC, DT, AdaBoost, and GradientBoosting, the performance is higher, up to 82.8% accuracy. However, there is still room for improvement with 82.8% accuracy, and more models such as XGB, RF, and LGBM can be introduced for performance comparison. *Modak & Jha (2023)* comprehensively evaluated the model performance on the Diabetic2

dataset using nine models: an integrated model, a linear model, and a nonlinear model. CatBoost performed the best, achieving 95.4% accuracy. The performance of the models on other datasets still needs to be evaluated. *Oliullah et al. (2024)* achieved 92.91% accuracy in predicting diabetes using an integrated model on the PIMA dataset. The introduction of Shapley enhances the interpretability of the machine learning model at the same time. However, all the machine learning classifiers compared are integrated models, and linear and nonlinear models can be added appropriately to compare the results. *Doğru, Buyrukoğlu & Arı(2023)* proposed a super learner model to determine cardinality as the optimal feature selection technique from five types of feature engineering. Then, three different datasets were used to measure the robustness of the proposed model, and 99.6%, 92%, and 98% accuracy were obtained, respectively. The study can appropriately increase the interpretability of the model. *Xu et al. (2022)* Prediction using clinical data from 2015–2021 in Chongqing tertiary hospital. LR, CART, XGB, and RF were used for diabetes prediction, where RF had the best performance with a diabetes classification accuracy of 73.5%. The study can be appropriate for adding more machine learning algorithms for prediction. At the same time, model accuracy needs to be improved. *Wu et al. (2022)* on the NHANES dataset, using RF classification accuracy of 92% accuracy, also increases the interpretability of the model by explaining the main features that affect diabetes prediction. However, the number of classifiers used in it is low, which can be increased appropriately for comparative analysis. *Liu et al. (2022)* trained using LR, DT, RF, and XGB models with the best accuracy of 75.03% for XGB model performance on the Wuhan City 2019–2020 follow-up dataset, which has 127,031 samples and a large sample size, also still with the use of Shapley additive interpretation (SHAP) to calculate and visualize feature importance. However, the study only used four machine learning models, which can be increased appropriately. *Qin et al. (2022)* used five machine learning classifiers for diabetes prediction on the NHANES data computer, of which the CATBoost classifier performed best, with an accuracy of 82.1%. Among the five machine learning models, dietary intake levels of energy, carbohydrates, and fat contributed the most to the prediction of diabetic patients. *Modak & Jha (2023)* used a range of machine learning techniques, including linear models, nonlinear models, integrated models, *etc.*, to predict diabetes on Kaggle's real-world dataset data machine. CatBoost performed the best, with an accuracy of 95.4%. However, the study was conducted only on a single dataset, and the robustness of the model can be subsequently verified on multiple datasets. *Amma (2024)* proposed the En-RfRsK model. The model consists of three machine-learning techniques: RF, SVM, and KNN. Experiments were conducted on the PIMA dataset with an accuracy of 88.89%. For the En-RfRsK model, the interpretability of the model can be increased appropriately, and the robustness of the model can be verified on multiple datasets. *Ahmed et al. (2022)* proposed SVM-ANN model with 94.87 prediction accuracy on the UCI Machine Learning Repository data machine. *Jiang et al. (2023)* designed a diabetes risk assessment model based on an RF classifier. The accuracy rate was 95.15% on the dataset of diabetic patients' follow-up records from 2016–2023 in Haizhu District, Guangzhou City, China. The study dataset has a large amount of data and a high accuracy rate. Still, the limitation is that the data features need to be more comprehensive, which may affect the accuracy of disease risk

prediction to some extent. *Jannoud et al. (2024)* proposed a hybrid multi-layer algorithm (MLHA) model. The first layer of the model consists of three different SVC, RF, and KNN. They operate in parallel and then output data. The output data is used as the input data for XGBoost and then added to the prediction. The accuracy is 86.5%, respectively.

In exploring the field of diabetes risk prediction, in addition to integrated learning as a technique, neural networks and deep learning algorithms have been used in academia. *Zhao et al. (2024)* conducted diabetes prediction experiments using clinical data from electronic data of patients attending the metabolic disease clinic at the Affiliated Hospital of Qingdao University. Five machine learning algorithms, artificial neural network (ANN), decision tree (DT), random forest (RF), support vector classification (SVC), and Gaussian Naive Bayes (GNB), were chosen to predict diabetes, and the results showed that ANN had the best performance, achieving an accuracy of 92.47%. This study can add more machine learning algorithms for diabetes prediction as appropriate. *Olisah, Smith & Smith (2022)* proposed a deep learning model, 2GDNN, for diabetes mellitus classification using Spearman correlation and polynomial regression for feature selection and missing value interpolation, respectively. Experiments were carried out on the PIMA and LMCH data computers, with accuracy of 97.34% and 97.28% model performance, respectively. *Chowdhury, Ayon & Hossain (2024)* used four data enhancement techniques, SMOTE-N, SMOTE-Tomek, and SMOTE-ENN, to improve the performance of machine learning algorithms using data enhancement techniques on the BRFSS-2021 data computer. Gradient Boost performs the best with an AUC of 0.789. This study only used descriptive statistics to describe the effect of the four data enhancement techniques on model performance, and it is recommended to include judgemental statistics such as paired samples $t$-tests to comprehensively evaluate the impact of the four data enhancement techniques on model performance.

In addition, the use of data balancing techniques and feature engineering techniques can significantly improve the performance of machine learning models when it comes to diabetes prediction. *Uddin et al. (2024)* used the 2019 diabetes and PIMA datasets for the study, one balanced and the other imbalanced. The imbalanced dataset was processed using the SMOTE technique. Then six machine learning models, logistic regression (LR), k-nearest neighbor, naive Bayes, RF, SVC, and DT, were selected for experimentation, which showed 97% accuracy on the 2019 diabetes dataset and 80% on the PIMA dataset. The results show that balancing the dataset significantly reduces the number of false negative tests. *Bhat, Ansari & Ansari (2024)* explored the effect of three feature engineering methods, CFS, SFS, and information gain, on the performance of machine learning models on the T2DM dataset. Diabetes prediction is done using six machine learning models: LR, SVC, GNB, DT, RF, and KNN. Finally, the best DT performance is 96.10%. The machine learning algorithms in this study are less and should be increased appropriately. *Waqas Khan et al. (2024)* used Chi-square, SFS, and mutual information for feature selection on the PIMA and Early Risk Diabetes datasets and RF, GB, ANN, Tab-Net, and SVC for predicting diabetes. RF and Tab-Net achieved 99.35% and 99.36% accuracy, respectively, and excellent model performance. *Shaukat et al. (2023)* selected KNN, RF, SVC, and LR classifiers and experimented with WEKA 3.8.1 and Python 3.10. Finally, the LR classifier

performed best with 81% accuracy. As appropriate, this experiment can be predicted by adding integrated models like XGB, light gradient-boosting machine, bagging, *etc.*

Summarising the existing studies, it is found that most of the current research strategies chosen by authors for diabetes research are integrated models, neural network and deep learning models, data balancing techniques, and feature engineering to improve prediction accuracy, with the use of integrated learning models accounting for a higher percentage. The vast majority of authors choose PIMA, NHANES, *etc.* as the training datasets, which are the classic datasets for diabetes research, but the sample size is small. Some authors also choose the dataset of hospital patients, which is characterised by a large sample size and wide age range, and the collection of patient data takes a long time. Current research has some shortcomings in the prediction of diabetes mellitus, mainly due to an overconcentration on the prediction of the population as a whole without giving enough attention to high-risk groups, such as middle-aged and older adults. In addition, most researchers have chosen a small number of control classifiers for classification prediction of diabetes mellitus, which limits the comprehensive assessment and demonstration of the performance of the classifiers. Relevant presentations are summarised in Table 1.

## MATERIALS & METHODS

The Diabetes Simple Diagnosis (*Kaggle, 2024*) and Diabetes Prediction Dataset (*Kaggle, 2023*) datasets from the Kaggle website were selected for testing. The data were first preprocessed, including solo heat coding, outlier removal, and age screening. The dataset was categorised into age groups: 50–60, 60–70, and 70–80. The dataset was then balanced using a few class samples synthesised using the SMOTE technique.

Then, it is trained using the Bagging-DT-RF integrated model and RF, GNB, LR, KNN, DT, ET, Adaboost, XGB, and 8 machine learning classifiers as controls. Finally, the performance of each classifier is compared using metrics such as accuracy, precision, recall, AUC, and $F1$ score. The research roadmap is shown in Fig. 1.

### Data

Two datasets, the Diabetes Simple Diagnosis and Diabetes Prediction datasets, were used for diabetes prediction in this study. The Diabetes Simple Diagnosis dataset consists of 88,380 samples with seven input features, of which 35,892 are over 50 years of age.

The target attributes were binary and included 8424 diabetic and 79,956 non-diabetic patients, with 6696 diabetic and 29,196 non-diabetic patients in the 50+ sample. For detailed data information, please refer to Table 2.

The Diabetes Prediction Dataset consists of 100,000 samples of eight input features, with 39,918 samples over 50 years old. The target attributes are binary and include 8,500 diabetic and 91,500 non-diabetic patients, with 6790 diabetic and 33,127 non-diabetic patients in the 50+ sample. Please refer to Table 3 for detailed data information.

### Data preprocessing

The input categorical variables were first preprocessed, *i.e.,* they were subjected to one-hot encoding. The gender variable on the Diabetes Simple Diagnosis dataset and the gender

**Table 1 Summary of relevant studies.**

| | | Data set | Methodologies | Vintage | Drawbacks | Performances |
|---|---|---|---|---|---|---|
| 1 | *Tripathi et al. (2023)* | PIMA | Ensemble Methods— Soft voting | Predictions were made using the integrated model in conjunction with soft voting, while performance was compared with six machine learning models, with higher performance reaching 82.8% accuracy. | The accuracy of 82.8% is relatively low; while the PIMA dataset samples are all female, there are fewer machine learning classifiers for comparison, and XGB, R, and LGBM models can be added appropriately for contrast. | 82.8% |
| 2 | *Modak & Jha (2023)* | Diabetic2 | CatBoost | Multiple machine learning models, such as LR, KNN, XGB, LGBM, and CatBoost, are used to predict heart disease, and various metrics are used to measure model performance. | Only one dataset, Diabetic2, was selected for training and testing, and the model's performance on other diabetes datasets was not tested. | 95.4% |
| 3 | *Uddin et al. (2024)* | 2019 Diabetes Dataset, PIMA | SMOTE-RF | SMOTE-RF achieved 97% and 80% accuracy.<br><br>Exploring the effect of balanced and unbalanced datasets on the experimental results showed that balanced datasets can significantly reduce the number of false negative tests. | A detailed explanation of how the SMOTE technique impacts the performance of machine learning models is not provided. | 97%, 80%. |
| 4 | *Oliullah et al. (2024)* | PIMA | Ensemble model | Diabetes prediction using integrated models and selecting XGB, Bagging, LGBM, and AdaBoost as base models achieved 92.91% accuracy.<br><br>The introduction of Shapley facilitated the interpretation of machine learning models. | All the machine learning classifiers compared are integrated models, and linear and non-linear models can be added as appropriate to compare results. | 92.91% |
| 5 | *Zhao et al. (2024)* | Electronic data of patients in the Metabolic Disease Specialist Clinic of the Affiliated Hospital of Qingdao University | ANN | The selected dataset is from accurate clinical data, which is authentic and reliable. The ANN classifier performance is excellent, reaching 92.47% accuracy. | Only five machine learning algorithms, ANN, DT, RF, SVC, and GNB, were selected to predict diabetes; more machine learning algorithms can be added appropriately. | 92.47% |

| | | Data set | Methodologies | Vintage | Drawbacks | Performances |
|---|---|---|---|---|---|---|
| 6 | *Bhat, Ansari & Ansari (2024)* | T2DM | Feature engineering-DT | To explore the effect of three feature engineering methods, CFS, SFS, and Information Gain, on the performance of machine learning models. Predictions are made using six machine learning models, namely LR, SVC, GNB, DT, RF, and KNN, with the final best DT performance of 96.10%. | More machine learning algorithms can be added for prediction as appropriate. | 96.10% |
| 7 | *Doğru, Buyrukoğlu & Arı(2023)* | Early Diabetes Risk Prediction, PIMA, Diabetes 130-US Hospitals | The hybrid super ensemble learning model | A super learner model is proposed, and three different datasets are used to measure its robustness. The proposed model obtained 99.6%, 92%, and 98% accuracy, respectively. Determine the cardinality as the optimal feature selection technique from five types of feature engineering. | The interpretability of the model can be increased appropriately. | 99.6%, 92%, 98% |
| 8 | *Xu et al. (2022)* | 2015–2021 Chongqing Tertiary Hospital Data | RF | Prediction using hospital clinical data. Diabetes prediction was made using LR, CART, XGB, RF, *etc.*, where RF performed the best with a diabetes classification accuracy of 73.5%. | More machine learning algorithms could be added for prediction as appropriate. Model accuracy needs to be improved. | 81.4% |
| 9 | *Wu et al. (2022)* | NHANES | RF | RF classification was 92% accurate with high accuracy. Increased the interpretability of the model to recognize the most critical risk factors associated with diabetes. | More machine learning algorithms can be added for prediction as appropriate. | 92% |

**Table 1** (*continued*)

|  |  | Data set | Methodologies | Vintage | Drawbacks | Performances |
|---|---|---|---|---|---|---|
| 10 | *Liu et al. (2022)* | 2019–2020 follow-up dataset | XGBoost | On the Wuhan City 2019–2020 follow-up dataset, LR, DT, RF, and XGB models were used for training, and the XGB model performed with the best accuracy of 75.03%.<br><br>Feature significance was calculated and visualized using Shapley additive interpretation (SHAP). | Only four machine learning models are used, and additional machine learning models can be added as appropriate. | 75.03% |
| 11 | *Waqas Khan et al. (2024)* | PIMA, Early Risk Diabetes Dataset | RF, Tab-Net | The study used Chi-square, SFS, and mutual information for feature selection and used RF, GB, ANN, Tab-Net, and SVC to predict diabetes mellitus. RF and Tab-Net achieved 99.35% and 99.36% accuracy, respectively. | The machine learning used in this study uses fewer machine learning models, and more machine learning classifiers could be added for control. | 99.35%, 99.36% |
| 12 | *Qin et al. (2022)* | NHANES | CATBoost | CATBoost classifier, which had an accuracy of 82.1%.<br><br>Among the five machine learning models, the dietary intake levels of energy, carbohydrates, and fats contributed the most to the prediction of diabetic patients. | Prediction accuracy needs to be improved.<br><br>Fewer of its learning models are used. | 82.1% |
| 13 | *Olisah, Smith & Smith (2022)* | PIMA, LMCH | 2GDNN | Spearman correlation and polynomial regression are used for feature selection and missing value interpolation.<br><br>Proposed deep learning model 2GDNN model for diabetes classification | The interpretability of the model can be increased appropriately. | 97.34%, 97.28% |
| 14 | *Modak & Jha (2023)* | Kaggle's Real World Dataset | CatBoost | Using a range of machine learning techniques, including linear models, non-linear models, integrated models, *etc.*, CatBoost performed best with an accuracy of 95.4%. | The robustness of the model can be verified on multiple datasets. | 95.4% |

**Table 1 (*continued*)**

| | | Data set | Methodologies | Vintage | Drawbacks | Performances |
|---|---|---|---|---|---|---|
| 15 | *Amma (2024)* | PIMA | En-RfRsK | The En-RfRsK model, which consists of three machine learning techniques: RF, SVM, and KNN, is proposed. | The interpretability of the model can be increased appropriately. | 88.89%. |
| | | | | The En-RfRSK method then obtained an accuracy of 88.89% on the PIMA data machine. | The robustness of the model can be verified on multiple datasets. | |
| 16 | *Ahmed et al. (2022)* | UCI Machine Learning Repository | SVM-ANN | The prediction accuracy of the proposed SVM-ANN model is 94.87. | The interpretability of the model can be increased appropriately. | 94.87% |
| 17 | *Shaukat et al. (2023)* and *Waqas Khan et al. (2024)* | PIMA | LR | KNN, RF, SVC, and LR classifiers were selected, and experiments were carried out using WEKA 3.8.1 and Python 3.10. finally, the LR classifier performed the best with 81% accuracy. | Integrated models such as XGB, LGBM, and Bagging can be added for prediction as appropriate. | 81% |
| 18 | *Jiang et al. (2023)* | Records of diabetic patients' follow-up in Haizhu District, Guangzhou City, China, 2016-2023, 252176 records | RF | A diabetes risk assessment model based on an RF classifier was designed. The accuracy was 95.15 percent. | The limitation of less comprehensive data characterization may somewhat affect the accuracy of disease risk prediction. | 95.15% |
| | | | | A method applicable to community-based mass screening for diabetes risk is provided. | | |
| 19 | *Jannoud et al. (2024)* | PIMA | Hybrid multi-layer algorithm (MLHA) | The first layer consists of three different SVC, RF, and KNN. They operate in parallel and then output the data; the output data is used as input data for XGBoost and then into prediction. The accuracy is 86.5%, respectively. | Only classifiers like RF, LR, SVC, and ANN were selected for this experiment to compare results. | 86.5% |

**Table 1** (*continued*)

|  |  | Data set | Methodologies | Vintage | Drawbacks | Performances |
|---|---|---|---|---|---|---|
| 20 | *Chowdhury, Ayon & Hossain (2024)* | BRFSS-2021 | Gradient Boost | Four data enhancement techniques, SMOTE-N, SMOTE-Tomek, and SMOTE-ENN, are used to improve the performance of machine learning algorithms using data enhancement techniques. Gradient Boost has the best performance, with an AUC of 0.789. | Only descriptive statistics were used to describe the effects of the four data enhancement techniques on model performance, and it is recommended that judgment line statistics such as paired samples t-tests be added to comprehensively evaluate the effects of the four data enhancement techniques on model performance. | 0.789 (AUC) |

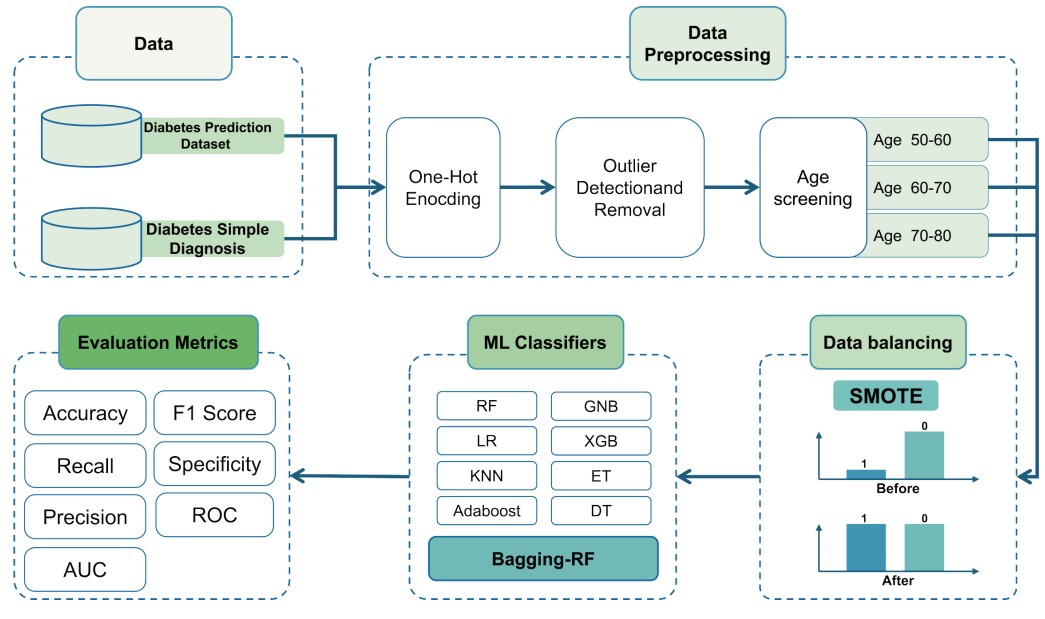

**Figure 1** **Flow chart of the study.**

and smoking history variables on the Diabetes Prediction Dataset were subjected to the one-hot encoding process to turn them into 0s and 1s.

In the outlier detection and removal stage, outliers that are beyond the range of plus or minus two standard deviations from the mean are removed, which can effectively reduce the impact of noise in the data on the performance of the machine learning model.

Outliers were detected for BMI, HbA1c level, and blood glucose level variables in the Diabetes Simple Diagnosis dataset. The mean and standard deviation were calculated first, then outliers outside the range of plus or minus 2 standard deviations from the mean were removed as shown in Eqs. (1) and (2). The mean BMI was 27.07 with a standard deviation of 13.22, and samples less than 13.22 and more excellent than 40.92 were removed. The

**Table 2  Diabetes simple diagnosis dataset details.**

|  | Feature | Description | Numerical value |
|---|---|---|---|
| F1 | Age | Age of the sample | 0–80 |
| F2 | Gender | Gender refers to the biological sex of the individual | Female, Male |
| F3 | BMl | Body Mass Index | 10–95 |
| F4 | High BP | Whether suffering from high blood pressure, 1 means yes, 0 means no 0. | 0,1 |
| F5 | FBS | Fasting blood glucose levels after overnight. | 80–300 |
| F6 | HbA1c level | Indicators of the average blood glucose level of the sample over the past 2–3 months. | 3.5–12 |
| F7 | Smoking | Smoking, 1 means yes, 0 means no. | 0, 1 |
| F8 | Diagnosis | Whether suffering from diabetes, 1 means suffering from diabetes, 0 means no. | 0, 1 |

**Table 3  Diabetes prediction dataset details.**

|  | Feature | Description | Numerical value |
|---|---|---|---|
| F1 | Gender | Gender refers to the biological sex of the individual | Female, Male |
| F2 | Age | Age of the sample | 0–80 |
| F3 | Hypertension | Whether the sample has hypertension, 1 means yes, 0 means no. | 0, 1 |
| F4 | Heart disease | Whether suffering from heart disease, 1 means suffering from heart disease, 0 means no. | 0,1 |
| F5 | Smoking history | Smoking history | Current, No Info, Ever, Former, Never, Not current |
| F6 | BMI | Body Mass Index | 10–96 |
| F7 | HbA1c level | Indicators of the average blood glucose level of the sample over the past 2–3 months. | 3.5–9 |
| F8 | Blood glucose level | Glucose content in blood. | 80–300 |
| F9 | Diabetes | Whether suffering from diabetes, 1 means suffering from diabetes, 0 means no. | 0, 1 |

mean HbA1c level was 5.53 with a standard deviation of 1.07; samples less than 3.39 and more significant than 7.68 were removed. The mean of FBS was 137.05 with a standard deviation of 1.07, and samples less than 3.39 and more significant than 7.68 were removed. The mean value for FBS was 137.05 with a standard deviation of 39.63, deleting samples less than 57.79 and greater than 216.31.

Outlier tests were performed on the BMI, HbA1c level, and blood glucose level variables in the Diabetes Prediction Dataset. The mean BMI was 28.93 with a standard deviation of 5.77, and samples smaller than 17.40 and larger than 40.46 were removed. The mean HbA1c level was 5.64 with a standard deviation of 1.14, and samples smaller than 3.36 and more significant than 7.92 were removed. The mean value of the HbA1c level was 5.64 with a standard deviation of 1.14, and samples less than 3.36 and more significant than 7.92 were removed. The mean blood glucose level was 140.87 with a standard deviation

**Table 4 Mean and standard deviation analysis table.**

|  | Mean ($\overline{x}$) | Standard deviation ($\sigma$) | Lower bound | Upper bound |
|---|---|---|---|---|
|  | Diabetes simple diagnosis | | | |
| BMI | 27.07 | 13.22 | 13.22 | 40.92 |
| HbA1c level | 5.53 | 1.07 | 3.39 | 7.68 |
| FBS | 137.05 | 39.63 | 57.79 | 216.31 |
|  | Diabetes prediction dataset | | | |
| BMI | 28.93 | 5.77 | 17.40 | 40.46 |
| HbA1c level | 5.64 | 1.14 | 3.36 | 7.92 |
| Blood glucose level | 140.87 | 43.32 | 54.24 | 227.50 |

of 43.32, and samples less than 54.24 and more excellent than 227.50 were removed. The table of means and standard deviations is shown in Table 4.

$$\text{Lower bound} = \overline{x} - 2 \times \sigma \tag{1}$$

$$\text{Upper bound} = \overline{x} + 2 \times \sigma. \tag{2}$$

The test set was age-stratified into three specific age groups:

- Age 50–60: between the ages of 50 and 60, in the transition from late middle age to early old age.
- Age 60–70: between the ages of 60 and 70, in the early stages of old age, this age group may face more age-related health challenges, such as chronic diseases and a gradual decline in physical functioning.
- Age 70–80: between the ages of 70 and 80, the advanced stage of ageing, individuals in this age group focus on managing chronic disease and maintaining quality of life.

This stratification strategy aims to provide insight into the performance and applicability of machine learning classifiers in different age groups of the senior population. By subdividing the test set into these specific age intervals, the classifiers' diagnostic accuracy and predictive power can be more precisely assessed across age groups.

## Data balancing processing

Unbalanced data distribution may negatively affect the predictive performance of machine learning models. To improve the generalisation ability and accuracy of the model, an SMOTE strategy is adopted to balance the dataset and solve the problem of unbalanced data distribution. SMOTE increases the number of minority class samples by generating synthetic data points between the minority class samples, which reduces the degree of unbalance in the dataset (*Arafa et al., 2022*).

The mechanism of action for generating synthetic samples in SMOTE is to create new data points by linearly interpolating the space between a small number of class samples and their immediate neighbours.

Firstly, identify the classes that have a small number of samples in the classification problem. After that, the nearest neighbours of each minority class sample were found

using Euclidean distance. These nearest neighbours should also be minority-class samples to ensure the relevance and quality of the synthetic samples. Then, $n$ samples ($n$ less than or equal to $k$) are randomly selected from the $k$ nearest neighbours of each minority class sample, and then linear interpolation is performed between the original sample and these selected nearest neighbours; for each selected nearest neighbour sample $xn$, the vector difference between the original sample and it is computed and then multiplied by a random number between 0 and 1 $\delta$, which determines the correlation of the newly synthesised sample xsmote with the vector difference in the original sample $x$ and the relative position between the nearest neighbours $xn$. The calculation formula is given in Eq. (3).

$$x_{\text{smote}} = x + \delta \times (x_n - x). \tag{3}$$

There are many other variants of the SMOTE technique from which it is derived, such as Borderline-SMOTE, ADASYN, SMOTE-Tomek, and so on (*Li et al., 2022*; *Mostafaei, Ahmadi & Shahrabi, 2023*; *Munshi, 2024*). With these methods, data from different age groups can be better balanced to optimise model training and prediction.

Looking at the data distribution for age 50–60, age 60–70, and age 70–80 on both datasets, it was observed that the data distribution needed to be balanced among the three age groups. Before the balancing process of the dataset, it was observed that the ratio of people without diabetes to people with diabetes was 10:1 in the six datasets, and by applying the SMOTE technique, the data distribution was adjusted so that the ratio of people without diabetes to people with diabetes reached a balanced state. The data distribution before and after balancing is shown in Fig. 2.

## Bagging-RF integrated learning classifier

A bagging classifier with DT as the base model is first configured. A grid search algorithm combined with 5-fold cross-validation is used to explore different hyperparameter combinations to optimise its performance systematically. The optimal parameter settings for the bagging classifier were determined through a grid search, including parameters such as the number of DTs, the size of each subsample, and the maximum depth of the decision tree. Using these optimal hyperparameters, the Bagging classifier was trained, and subsequently, its predictions on the training set were fed into the RF classifier as new training data.

Combining the advantages of the two integrated learning methods improves the accuracy and stability of the prediction of unknown data. The training process of the Random Forest classifier utilises the output of the Bagging classifier, thus integrating more decision tree prediction information, with the expectation of achieving better performance in all types of classification evaluation metrics. The mechanism of bagging-RF action is shown in the pseudo-code below.

Figure 3 shows the Bagging-RF flowchart. This multi-level integrated learning strategy effectively incorporates the advantages of multiple models to achieve better classification results on complex datasets.

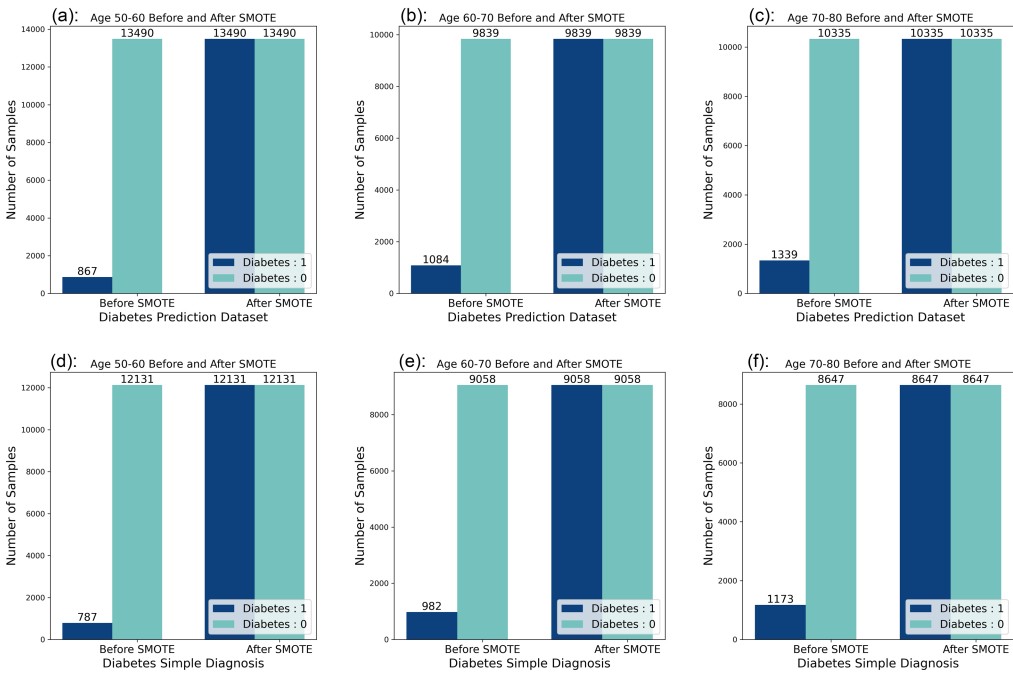

**Figure 2** **Data distribution before and after balancing.** (A) Diabetes Prediction Dataset 50–60 years of age. (B) Diabetes Prediction Dataset 60–70 years old. (C) Diabetes Prediction Dataset 70–80 years of age. (D) Diabetes Simple Diagnosis Dataset 50–60 years old. (E) Diabetes Simple Diagnosis 60–70 years dataset. (F) Diabetes Simple Diagnosis Dataset 70–80 years old.

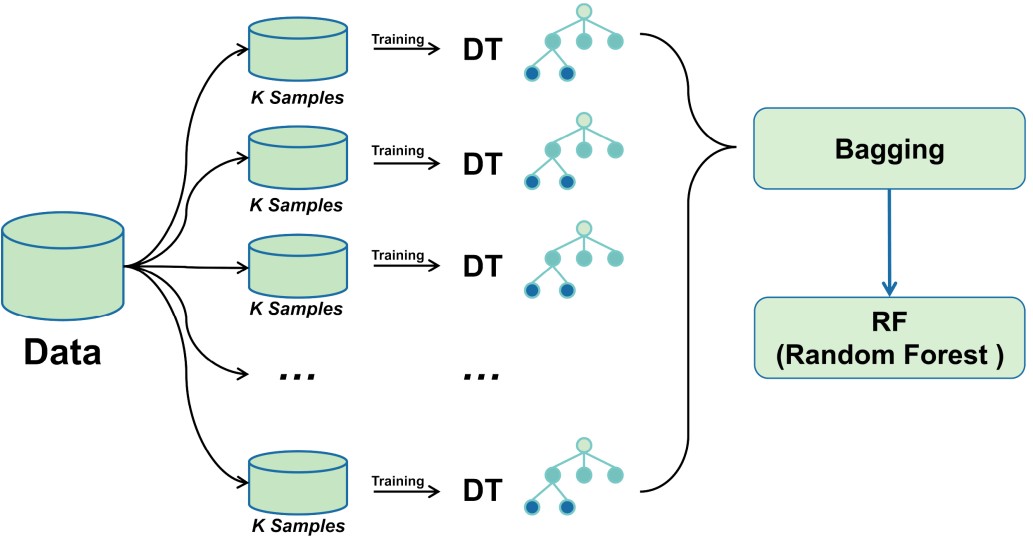

**Figure 3** **Bagging-RF flow chart.**

**Algorithm: Bagging-RF**
**Require**
   X training features, y training label
**Ensure**
   Trained Bagging-RF model and evaluation metrics

    **1. Importing the processed diabetes dataset:**
      X: Predictor variables indicative of diabetes mellitus
      y: Binary outcome indicating the presence or absence of diabetes mellitus

    **2. Split dataset into training and testing sets:**
      $X_{train}$, $X_{test}$, $y_{train}$, $y_{test} \leftarrow$ train_test_split(X, y, test_size = 0.2, random_state = 42)

    **3. Train Bagging model:**
      Bagging_model $\leftarrow$ BaggingClassifier(random_state = 42)
      Bagging_model.fit($X_{train}$,$y_{train}$)
      Bagging_predictions $\leftarrow$ bagging_model.predict($X_{test}$)

    **4. Prepare new feature sets for Random Forest:**
      Bagging_predictions_train $\leftarrow$ bagging_model.predict($X_{train}$).reshape($-1,1$)
      Bagging_predictions_test $\leftarrow$ bagging_predictions.reshape($-1,1$)
      $X_{train\_rf} \leftarrow$ np.hstack(($X_{train}$, bagging_predictions_train))
      $X_{test\_rf} \leftarrow$ np.hstack(($X_{test}$, baggin_ predictions_test))

    **5. Train Random Forest model:**
      rf_model $\leftarrow$ RandomForestClassifier(n_estimators = 100,random_state = 42)

      rf_model.fit($X_{train\_rf}$, $y_{train}$)
      rf_predictions $\leftarrow$ rf_model.predict($X_{test\_rf}$)

    **6. Evaluate model performance:**
      accuracy $\leftarrow$ accuracy_score($y_{test}$, rf_predictions)
      precision $\leftarrow$ precision_score($y_{test}$, rf_predictions,)
      recall $\leftarrow$ recall_score($y_{test}$, rf_predictions)
      f1 $\leftarrow$ f1_score($y_{test}$, rf_predictions)
      conf_matrix $\leftarrow$ confusion matrix($y_{test}$,rf_predictions)
      TN $\leftarrow$conf_matrix[0,0]
      FN $\leftarrow$conf_matrix[1,0]
      FP $\leftarrow$conf_matrix[0,1]
      TP $\leftarrow$conf_matrix[1,1]

## Machine learning classifiers

This section describes eight machine learning classifiers that will be used for diabetes risk prediction in the middle-aged and elderly populations. The selected classifiers include GNB, KNN, LR, RF, ET, DT, AdaBoost, and XGB. These classifiers cover several different domains, such as integrated learning (*Buyrukoğlu & Savaş, 2022*; *Buyrukoğlu, 2021*), Bayesian methods, and linear and nonlinear models. They are intended to ensure

that this study is able to comprehensively assess the performance of the different types of classifiers on diabetes risk prediction. These eight classifier models are analysed for performance comparison against the Bagging-RF model as a control group. Next, the principles and applications of each classifier are briefly described.

K-nearest neighbors (KNN): The K-nearest neighbors algorithm is instance-based learning that predicts the category of a new data point by finding the k closest points in the dataset to the latest data point. This algorithm relies on a distance metric (*e.g.*, Euclidean distance). It assigns the new data point category based on majority voting on the categories of its k nearest neighbours.

Logistic regression (LR): Logistic regression is a statistical model for solving binary classification problems. The model represents the likelihood that an outcome belongs to a particular category by applying a logistic function that transforms the output value of a linear regression into a probability value between 0 and 1. This method is applicable when the data set is linearly differentiable (*Biesheuvel et al., 2008*).

Random forest (RF): Random forest is an integrated learning model that constructs multiple decision trees and aggregates their predictions to improve prediction accuracy by averaging or majority voting. Random Forest increases the variability between models by using a different subset of data and a randomly selected subset of features for each decision tree, thus improving overall generalisation.

Extra trees classifier (ET): The extra trees classifier is a variant of random forest that employs more randomness in pruning decision trees. Specifically, extra trees creates more decision trees by randomly selecting the features to be segmented and the cut points to increase the integration's diversity further (*Geurts, Ernst & Wehenkel, 2006*).

Decision tree (DT): A decision tree classifier is a recursive tree structure used to partition data into different classes or make regression predictions based on the value of features. Each internal node represents a feature, each branch represents a decision rule, and the leaf nodes represent the final predicted output.

AdaBoost: The AdaBoost classifier is an adaptive boosting algorithm that integrates multiple weak classifiers into one robust classifier through a weighted voting mechanism. AdaBoost adaptively adjusts the weights of the training samples and the error rate of the weak classifiers during the training process so that in each iteration, samples that the previous classifiers have misclassified will be given higher weights, thus forcing the subsequent classifiers to pay more attention. Weights, thus forcing the subsequent classifiers to pay more attention.

XGBoost: The XGBoost classifier is an optimised distributed gradient-boosting library designed to achieve high-speed and high-performance gradient-boosting decision trees. XGBoost provides several tunable parameters to control the complexity of the model and prevent overfitting. It also uses techniques such as parallel processing and systematic optimisation to speed up the model's training. The classifier has achieved excellent results in many machine learning competitions (*Chen & Guestrin, 2016*).

GNB: A Gaussian naive Bayes classifier is a probabilistic-based classification technique that belongs to the class of simple Bayesian algorithms. This algorithm is particularly suitable for dealing with classification problems where the feature data exhibit Gaussian

**Table 5  Hyperparameter settings.**

| Classifier | Hyperparameters |
|---|---|
| GNB | var_smoothing: 1e−9 |
| DT | max_depth: [None, 10, 20, 30], min_samples_split: [2, 4, 6], min_samples_leaf: [1, 2, 4], criterion: [gini, entropy] |
| LR | C: [0.1, 1, 10, 100], penalty: [l1, l2], solver: [liblinear, saga] |
| KNN | n_neighbors: [3, 5, 7, 9], weights: [uniform, distance], metric: [euclidean, manhattan, minkowski] |
| ET | n_estimators: [50, 30, 10], max_depth: [None, 10, 20, 30], min_samples_split: [2, 4, 6], min_samples_leaf: [1, 2, 4], bootstrap: [True, False] |
| RF | n_estimators: [50, 30, 10], max_depth: [None, 10, 20, 30], min_samples_split: [2, 4, 6], min_samples_leaf: [1, 2, 4], bootstrap: [True, False] |
| Adaboost | n_estimators: [50, 30, 10], learning_rate: [1.0, 0.5, 0.1], algorithm: [SAMME, SAMME.R], base_estimator__max_depth: [1, 2, 3] |
| XGB | max_depth: [3, 4, 5], learning_rate: [0.01, 0.05, 0.1], n_estimators: [50, 30, 10], subsample: [0.7, 0.8, 0.9], colsample_bytree: [0.7, 0.8, 0.9] |
| Bagging-RF | n_estimators: [10, 50, 100, 200], max_samples: [0.5, 0.7, 1.0], base_estimator__max_depth: [None, 10, 20, 30], bootstrap: [True, False], oob_score: [True, False] |

distribution characteristics. Its strength lies in its simple and efficient model structure, which predicts the class attribution of the samples by calculating the Gaussian distribution parameters of the features. This algorithm demonstrates excellent performance and scalability when dealing with large-scale datasets (*Alanazi & Alanazi, 2024*).

## Grid search

Optimising the performance of classifiers often involves adjusting the model's hyperparameters. Grid search is a commonly used method for hyperparameter optimisation that searches for the optimal combination of parameters by traversing a given grid of parameters. In this study, 5K cross-validation using grid search classes was used to optimise the hyperparameters of the machine learning classifier.

First, define the range of hyperparameter values for each classifier. The hyperparameter ranges are shown in Table 5. Launch the grid search algorithm to traverse each set of parameters in the grid and evaluate their performance through cross-validation. This experiment used a five-fold cross-validation strategy to divide the dataset into four parts. In each iteration, four parts of the data were selected as the training set, and the remaining parts were used as the validation set. 4/5 of the data were used for model training, and the remaining 1/5 was used as the validation set to compute the model's performance. Accuracy was chosen as a measure of model performance on the validation set. All possible combinations of parameters were evaluated using a grid search algorithm. Finally, based on the evaluation results, the best combination of hyperparameters is selected for each model.

The experimental setup details are shown in Table 6.

## Evaluation indicators

This section discusses the ML model evaluation metrics used in this study. We used accuracy, precision, recall, and $F1$ score as evaluation metrics for our ML model

| S.No. | Components | Detail |
|---|---|---|
| | Table 6  Experimental setup detail. | |
| 1 | Hardware | AMD R55600G |
| 2 | Operating system | Windows 11 |
| 3 | Primary storage | 16 GB RAM |
| 4 | Data file storage | MS Excel |
| 5 | Programming language | Python |
| 6 | Python required libraries | Pandas, numpy, sklearn, xgboost, time, seaborn, Matplotlib |
| 7 | IDE | Jupyter Notebook |

(*Naidu, Zuva & Sibanda, 2023*; *Panesar, 2021*; *Rainio, Teuho & Klén, 2024*). Below is a brief description of each evaluation metric and its significance in the classification of diabetes.

Accuracy: Accuracy is the number of correctly predicted samples as a proportion of the total number of samples. In diabetes classification, accuracy provides an overall perspective, telling us the proportion of all test samples the model correctly classifies. A model with a high accuracy rate correctly distinguishes people with diabetes from non-diabetics in most cases. It is shown in Eq. (4), where TP (true positives) is the number of confirmed cases, *i.e.,* the number of samples correctly predicted by the model to be in the positive category, TN (true negatives) is the number of true negatives, *i.e.,* the number of samples correctly predicted by the model to be in the harmful category, FP (false positives) is the number of false positives, *i.e.,* the number of samples incorrectly predicted by the model to be in the positive category, and FN (false negatives) are false negatives, *i.e.,* the number of samples that the model incorrectly predicts as antagonistic classes (*Bingol et al., 2023*; *Kiziloluk et al., 2024*).

$$Accuracy = \frac{TP + TN}{TP + TN + FP + FN}. \tag{4}$$

Precision: Precision, also known as the accuracy of optimistic class predictions, measures the proportion of all samples predicted by the model to be in the positive class that are actually in the positive class and is an essential metric in the performance evaluation of classification models, especially in binary classification problems. In diabetes classification, accuracy is concerned with the probability that when the model predicts that an individual has diabetes, this prediction is accurate. In other words, it measures the proportion of all individuals diagnosed with diabetes who do have the disease. The formula for this is shown in Eq. (5).

$$Precision = \frac{TP}{TP + FP}. \tag{5}$$

Recall: Recall measures the proportion of all samples in the positive class that are correctly predicted to be in the positive class by the model. In diabetes classification, recall is concerned with the ability of the model to identify all actual diabetic patients. It represents the proportion of all diabetic patients that are correctly diagnosed. A high recall rate implies a low rate of missed diagnoses, which is essential in areas such as medical

diagnosis, where missed diagnoses may lead to untreated diseases. Its formula is shown in Eq. (6).

$$\text{Recall} = \frac{TP}{TP + FN}. \tag{6}$$

*F*1 score: The *F*1 score is a classification model performance metric that combines precision and recall. It provides a single metric to evaluate a model's performance in terms of both precision and recall. The *F*1 score suffers when there is a large gap between precision and recall, forcing the model to strike a balance between the two metrics. Its formula is shown in Eq. (7).

$$F1 \text{ Score} = 2 \times \left( \frac{\text{Precision} \times \text{Recall}}{\text{Precision} + \text{Recall}} \right). \tag{7}$$

Specificity: Specificity measures a model's ability to correctly identify negative samples in binary classification, with higher values near 1 indicating better performance. It is vital in areas where false positives are detrimental, such as medical screening. Used alongside metrics like precision and *F*1 scores, specificity offers a broader evaluation of model effectiveness. Its formula is commonly presented with recall to assess a model's performance on both classes. The formula is shown in Eq. (8).

$$\text{Specificity} = \frac{TN}{TN + FP}. \tag{8}$$

ROC curve: The ROC curve evaluates the performance of a binary classification model. It demonstrates the model's ability to identify positive classes by plotting the relationship between the recall and the value of 1 minus specificity at different thresholds. The closer the curve is to the upper left corner, the better the model performance is, while the area under the curve (AUC) quantifies the overall effectiveness of the model, and the closer the AUC value is to 1, the better the model's classification ability is.

## RESULTS

### Experimental results on the diabetes prediction dataset

Table 7 and Fig. 4 show the prediction results of each machine learning model on the Diabetes Prediction Dataset dataset before and after data balancing for three age groups: 50–60 years, 60–70 years, and 70–80 years. Figure 4 is divided into six subfigures, a, b, c, d, e, and f, which show the prediction analysis results before and after the data balancing process for three different age groups (50–60 years old, 60–70 years old, and 70–80 years old). Subfigures a, c, and e correspond to the prediction results for ages 50–60, 60–70, and 70–80 before data balancing, while subfigures b, d, and f correspond to the prediction results for ages 50–60, 60–70, and 70–80 after data balancing.

Referring to Table 7 and Fig. 4, the Bagging-RF classifier before data balancing achieved 96.59% and 95.94% accuracy and *F*1-Score for diabetes detection in the 50–60-year-olds; 93.41% and 91.90% accuracy and *F*1-Score in the 60–70-year-olds; 91.69% and 89.83% accuracy and *F*1-Score in the 70–80-year-olds; and 91.69% and 89.83% performance of the integrated learning model ET, RF, AdaBoost, and XGB classifiers. In addition, the

**Table 7  Experimental data table for diabetes prediction dataset dataset.**

| ML | Data balancing | | | | | No data balancing | | | | |
|---|---|---|---|---|---|---|---|---|---|---|
| ML | Accuracy | F1 score | Specificity | Precision | Recall | Accuracy | F1 score | Specificity | Precision | Recall |
| | | | | | Age 50–60 | | | | | |
| GNB | 73.98% | 72.27% | 49.1% | 82.4% | 74.0% | 79.42% | 84.40% | 79.9% | 92.9% | 79.4% |
| DT | 95.94% | 95.94% | 96.3% | 95.9% | 95.9% | 95.86% | 95.10% | 99.6% | 95.5% | 95.9% |
| LR | 90.55% | 90.53% | 94.4% | 90.8% | 90.6% | 94.92% | 93.68% | 99.5% | 94.1% | 94.9% |
| KNN | 95.33% | 95.33% | 91.7% | 95.6% | 95.3% | 94.74% | 92.86% | 100.0% | 94.8% | 94.7% |
| ET | 97.57% | 97.57% | 98.5% | 97.6% | 97.6% | 95.33% | 94.04% | 99.9% | 95.3% | 95.3% |
| RF | 97.35% | 97.35% | 98.8% | 97.4% | 97.4% | 96.03% | 95.16% | 99.9% | 96.1% | 96.0% |
| ADB | 97.33% | 97.33% | 100.0% | 97.5% | 97.3% | 96.07% | 95.17% | 100.0% | 96.2% | 96.1% |
| XGB | 95.96% | 95.96% | 98.2% | 96.1% | 96.0% | 96.10% | 95.22% | 100.0% | 96.3% | 96.1% |
| Bagging-RF | 97.35% | 97.35% | 98.4% | 97.4% | 97.3% | 96.59% | 95.94% | 99.9% | 96.6% | 96.6% |
| | | | | | Age 60–70 | | | | | |
| GNB | 80.89% | 80.87% | 77.0% | 81.2% | 80.9% | 87.96% | 87.98% | 93.4% | 88.0% | 88.0% |
| DT | 93.47% | 93.47% | 94.5% | 93.5% | 93.5% | 93.09% | 91.90% | 98.9% | 92.2% | 93.1% |
| LR | 88.14% | 88.11% | 91.8% | 88.3% | 88.1% | 91.44% | 89.46% | 98.6% | 89.4% | 91.4% |
| KNN | 93.14% | 93.13% | 89.0% | 93.5% | 93.1% | 91.53% | 88.97% | 99.3% | 89.6% | 91.5% |
| ET | 95.58% | 95.58% | 97.3% | 95.6% | 95.6% | 92.91% | 91.27% | 99.4% | 92.2% | 92.9% |
| RF | 95.60% | 95.60% | 97.5% | 95.7% | 95.6% | 93.64% | 92.10% | 99.9% | 93.8% | 93.6% |
| ADB | 95.53% | 95.52% | 99.9% | 95.9% | 95.5% | 93.41% | 91.94% | 99.6% | 93.1% | 93.4% |
| XGB | 94.13% | 94.13% | 95.2% | 94.2% | 94.1% | 93.50% | 91.98% | 99.8% | 93.4% | 93.5% |
| Bagging-RF | 95.55% | 95.55% | 97.1% | 95.6% | 95.6% | 93.41% | 91.90% | 99.6% | 93.2% | 93.4% |
| | | | | | Age 70–80 | | | | | |
| GNB | 79.00% | 79.00% | 76.6% | 79.1% | 79.0% | 84.58% | 84.53% | 91.4% | 84.5% | 84.6% |
| DT | 94.61% | 94.60% | 95.4% | 94.6% | 94.6% | 91.82% | 90.63% | 98.6% | 91.1% | 91.8% |
| LR | 86.72% | 86.71% | 89.1% | 86.8% | 86.7% | 90.02% | 88.25% | 98.1% | 88.5% | 90.0% |
| KNN | 92.21% | 92.20% | 87.5% | 92.6% | 92.2% | 89.25% | 86.42% | 98.8% | 87.2% | 89.3% |
| ET | 94.36% | 94.36% | 95.6% | 94.4% | 94.4% | 92.25% | 90.72% | 99.7% | 92.4% | 92.3% |
| RF | 94.97% | 94.97% | 96.8% | 95.0% | 95.0% | 92.63% | 91.14% | 100.0% | 93.1% | 92.6% |
| ADB | 94.90% | 94.88% | 99.8% | 95.3% | 94.9% | 92.63% | 91.17% | 99.9% | 93.1% | 92.6% |
| XGB | 93.78% | 93.78% | 95.2% | 93.8% | 93.8% | 92.42% | 91.05% | 99.5% | 92.4% | 92.4% |
| Bagging-RF | 95.14% | 95.13% | 97.7% | 95.3% | 95.1% | 91.69% | 89.83% | 100.0% | 92.3% | 91.7% |

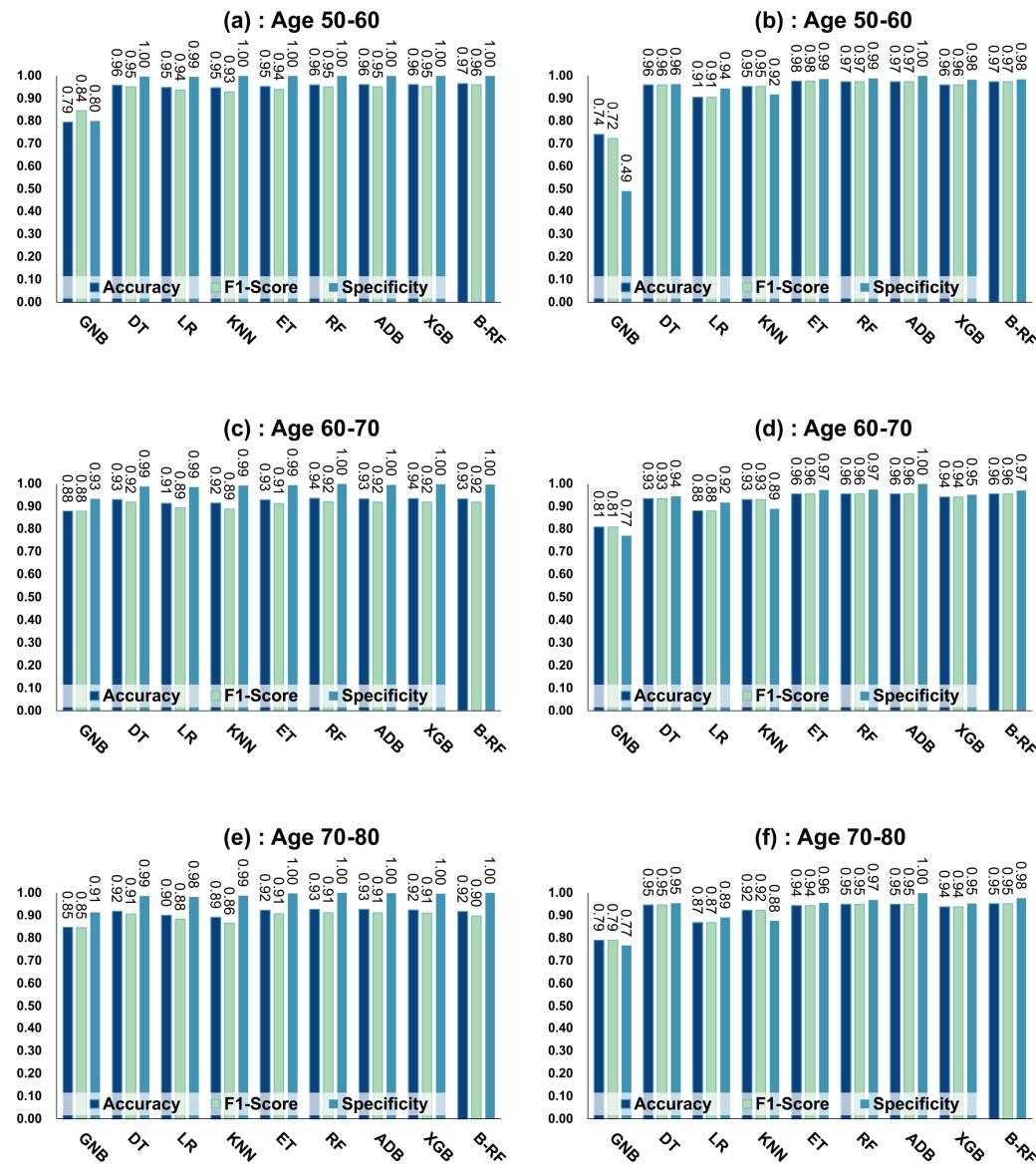

**Figure 4 Experimental chart of diabetes prediction dataset.** (A) Age 50–60 groups without balanced data sets. (B) Age 50–60 groups balanced data sets. (C) Age 60–70 groups without balanced data sets. (D) Age 60–70 groups balanced data sets. (E) Age 70–80 groups without balanced data sets. (F) Age 70–80 groups balanced data sets. Note: Bagging-RF is abbreviated to B-RF.

integrated learning models ET, RF, AdaBoost, and XGB classifiers all perform well. It is worth mentioning that on the dataset before data balancing, all machine learning classifiers' specificity is higher than 98% except the GNB classifier.

After the data balancing process, the performance of the Bagging-RF classifier was improved, achieving 97.35% and 97.35% accuracy and $F1$-score in the detection of diabetes in the 50–60-year-old population and 95.55% and 95.55% accuracy and $F1$-score in the 60–70-year-old population sample. The accuracy and $F1$-score of 94.61% and

94.60% were achieved in the sample of the 70–80-year-old population. After the data balancing process, the performance of all the models except the LR classifier was improved, and the accuracy of the integrated learning models, ET, RF, AdaBoost, and XGB, were all kept above 94%. Furthermore, the specificity of all classifiers decreased to varying degrees following the data balancing process, which could be due to a reduction in the proportion of non-diseased samples in the overall population, weakening the model's ability to recognise such categories.

Compared with the eight base classifiers, the Bagging-RF integrated learning classifier demonstrates optimal performance before and after the data balancing process. The confusion matrix of the Bagging-RF model is shown in Fig. 5. It is worth noting that the overall performance of the RF, AdaBoost, and XGB classifiers before and after the data balancing process, although not as good as that of the Bagging-RF integrated learning classifier, is always tiny, which shows the excellent performance of the integrated learning classifier. The ROC curves are shown in Fig. 6. By observing the ROC curves of each model, it is found that the AUC values of each model are significantly improved after data balancing treatment, among which the Bagging-RF, ET, RF, AdaBoost, and XGB models after data balancing treatment are shown in Fig. 5. AdaBoost and XGB integrated models all perform well.

### Experimental results on diabetes simple diagnosis

Table 8 and Fig. 7 show the prediction results of each machine learning model on the Diabetes Simple Diagnosis dataset for the three age groups of 50–60, 60–70, and 70–80 years of age for the dataset before balancing *versus* after balancing the data. Figure 5 is divided into six subfigures, a, b, c, d, e, and f, which show the predictive analysis results for three different age groups (50–60, 60–70, and 70–80) before and after the data balancing process. Subfigures a, c, and e correspond to the predicted results for ages 50–60, 60–70, and 70–80 before data balancing, while subfigures b, d, and f correspond to the predicted results for ages 50–60, 60–70, and 70–80 after data balancing.

Referring to Table 8 and Fig. 7, before data balancing, the Bagging-RF classifier achieved 95.63% and 94.42% accuracy and $F1$-score for diabetes detection in the 50–60 year old population; 93.48% and 92.07% accuracy and $F1$-score in the 60–70 year old population sample; and 91.45% and 89.52% accuracy and $F1$-score in the 70–80 year old population sample. All the models except LR, KNN, and GNB were above 90% accuracy and $F1$-score. Accuracy and $F1$-Score of 91.45% and 89.52% were achieved in the population sample of 70–80 years old; accuracy and $F1$-Score of 93.48% and 92.07% were achieved in the population sample of 60–70 years old; and 91.45% and 89.52% were achieved in the population sample of 70–80 years old. Except for GNB, which exhibited a specificity of approximately 95%, the other models had specificity scores of greater than 98%.

After the data were balanced, the performance of the Bagging-RF classifier was improved, achieving 97.03% and 97.03% accuracy and $F1$-score for diabetes detection in the 50–60-year-olds; 94.90% and 94.89% accuracy and $F1$-score in the 60–70-year-olds; and 93.70% and 93.70% accuracy and $F1$-score in the 70–80-year-olds. Of the eight models after data

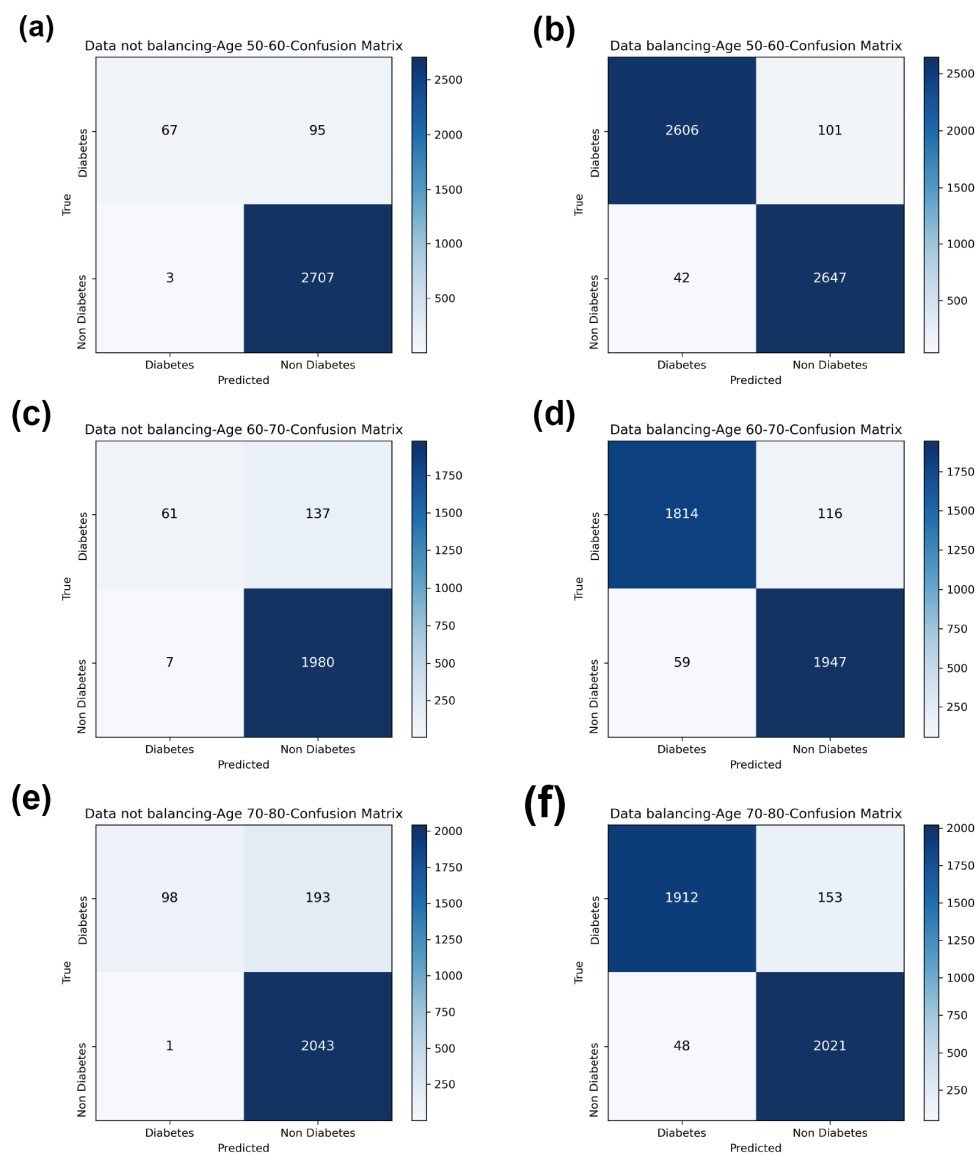

**Figure 5 Diabetes Prediction Dataset confusion matrix.** (A) Age 50–60 groups without balanced confusion matrix. (B) Age 50–60 groups balanced confusion matrix. (C) Age 60–70 groups without confusion matrix. (D) Age 60–70 groups balanced confusion matrix. (E) Age 70–80 groups without confusion matrix. (F) Age 70–80 groups balanced confusion matrix.

balancing, only the DT model reached 97.03% accuracy and $F1$-score in the 50–60-year-olds. 94.89%, 93.70%, and 93.70% for accuracy and $F1$-score in the sample of people aged 70–80. After data balancing, only the performance of the DT classifier was improved in the eight models, while the performance of the rest of the base classifiers decreased to different degrees. Additionally, the specificity scores of the models were decreased to varying degrees.

In comparison with the remaining eight models, the Bagging-RF integrated learning classifiers again show optimal performance, both before and after the data balancing

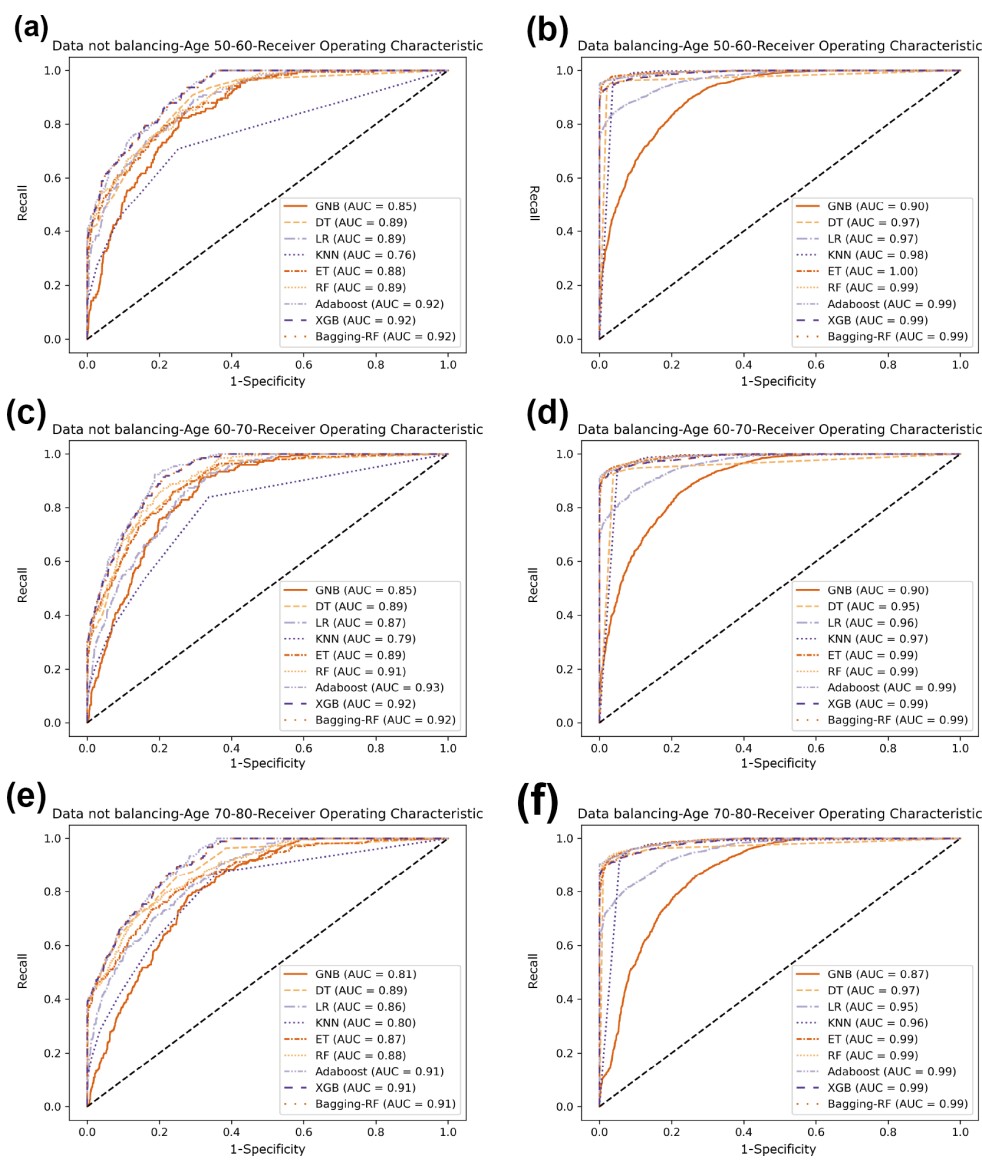

**Figure 6** **Diabetes Prediction Dataset ROC curve.** (A) Age 50–60 groups without a balanced ROC curve. (B) Age 50–60 groups balanced ROC curve. (C) Age 60–70 groups without a balanced ROC curve. (D) Age 60–70 groups balanced ROC curve. (E) Age 70–80 groups without a balanced ROC curve. (F) Age 70–80 groups balanced ROC curve.

process. The Bagging-RF model confusion matrix is shown in Fig. 8, and the ROC curves of each model are shown in Fig. 9. By observing the ROC curves of each model, it is found that after the data balancing process, the AUC values of all models are improved, among which the integrated models such as Bagging-RF, ET, RF, Adaboost, and XGB are all excellent. By observing the ROC curves of each model, it is found that the AUC values of each model are improved after the data balancing process, and the integrated models such as Bagging-RF, ET, RF, Adaboost, and XGB perform well after the data balancing process.

**Table 8** Experimental data table for Diabetes Simple Diagnosis dataset.

| ML | Data balancing | | | | | No data balancing | | | | |
|---|---|---|---|---|---|---|---|---|---|---|
| ML | Accuracy | F1 Score | Specificity | Precision | Recall | Accuracy | F1 Score | Specificity | Precision | Recall |
| | | | | | Age 50–60 | | | | | |
| GNB | 79.35% | 79.05% | 67.2% | 81.3% | 79.4% | 91.72% | 91.36% | 96.1% | 91.0% | 91.7% |
| DT | 96.04% | 96.04% | 96.2% | 96.0% | 96.0% | 95.01% | 94.05% | 99.2% | 94.1% | 95.0% |
| LR | 78.65% | 78.60% | 73.5% | 79.0% | 78.7% | 94.47% | 92.71% | 99.6% | 93.2% | 94.5% |
| KNN | 89.57% | 89.52% | 82.1% | 90.5% | 89.6% | 94.08% | 91.43% | 99.9% | 92.5% | 94.1% |
| ET | 93.55% | 93.55% | 91.5% | 93.6% | 93.6% | 95.47% | 94.19% | 100.0% | 95.6% | 95.5% |
| RF | 94.52% | 94.52% | 94.2% | 94.5% | 94.5% | 95.59% | 94.36% | 100.0% | 95.8% | 95.6% |
| ADB | 94.42% | 94.41% | 97.5% | 94.6% | 94.4% | 95.59% | 94.36% | 100.0% | 95.8% | 95.6% |
| XGB | 92.73% | 92.72% | 94.5% | 92.8% | 92.7% | 95.59% | 94.36% | 100.0% | 95.8% | 95.6% |
| Bagging-RF | 97.03% | 97.03% | 98.0% | 97.0% | 97.0% | 95.63% | 94.42% | 100.0% | 95.8% | 95.6% |
| | | | | | Age 60–70 | | | | | |
| GNB | 78.23% | 77.94% | 66.2% | 80.4% | 78.2% | 88.10% | 86.86% | 95.4% | 86.0% | 88.1% |
| DT | 94.54% | 94.54% | 94.5% | 94.5% | 94.5% | 92.63% | 91.63% | 98.4% | 91.7% | 92.6% |
| LR | 77.32% | 77.30% | 73.4% | 77.6% | 77.3% | 91.83% | 89.78% | 99.3% | 90.9% | 91.8% |
| KNN | 87.58% | 87.51% | 78.8% | 89.0% | 87.6% | 89.74% | 86.31% | 98.8% | 85.3% | 89.7% |
| ET | 91.42% | 91.42% | 88.2% | 91.6% | 91.4% | 93.33% | 91.90% | 99.7% | 93.3% | 93.3% |
| RF | 91.39% | 91.39% | 89.4% | 91.5% | 91.4% | 93.53% | 92.08% | 99.9% | 93.9% | 93.5% |
| ADB | 92.80% | 92.78% | 96.0% | 93.0% | 92.8% | 93.48% | 91.97% | 100.0% | 93.9% | 93.5% |
| XGB | 89.93% | 89.93% | 90.6% | 89.9% | 89.9% | 93.48% | 91.97% | 100.0% | 93.9% | 93.5% |
| Bagging-RF | 94.90% | 94.89% | 95.4% | 94.9% | 94.9% | 93.48% | 92.07% | 99.8% | 93.6% | 93.5% |
| | | | | | Age 70–80 | | | | | |
| GNB | 79.01% | 78.66% | 66.3% | 80.9% | 79.0% | 87.27% | 85.69% | 95.6% | 84.7% | 87.3% |
| DT | 92.74% | 92.74% | 93.5% | 92.8% | 92.7% | 91.14% | 89.47% | 98.7% | 90.1% | 91.1% |
| LR | 76.41% | 76.39% | 73.5% | 76.5% | 76.4% | 89.15% | 86.94% | 97.8% | 86.7% | 89.2% |
| KNN | 86.30% | 86.19% | 77.9% | 87.3% | 86.3% | 88.49% | 84.99% | 98.6% | 84.6% | 88.5% |
| ET | 89.74% | 89.74% | 89.0% | 89.7% | 89.7% | 91.40% | 89.23% | 99.7% | 91.3% | 91.4% |
| RF | 91.62% | 91.61% | 89.5% | 91.7% | 91.6% | 91.65% | 89.48% | 99.9% | 92.1% | 91.7% |
| ADB | 91.36% | 91.33% | 97.7% | 92.1% | 91.4% | 91.70% | 89.61% | 99.8% | 92.0% | 91.7% |
| XGB | 90.43% | 90.43% | 92.9% | 90.5% | 90.4% | 91.70% | 89.57% | 99.9% | 92.1% | 91.7% |
| Bagging-RF | 93.70% | 93.70% | 94.9% | 93.7% | 93.7% | 91.45% | 89.52% | 99.4% | 91.0% | 91.4% |

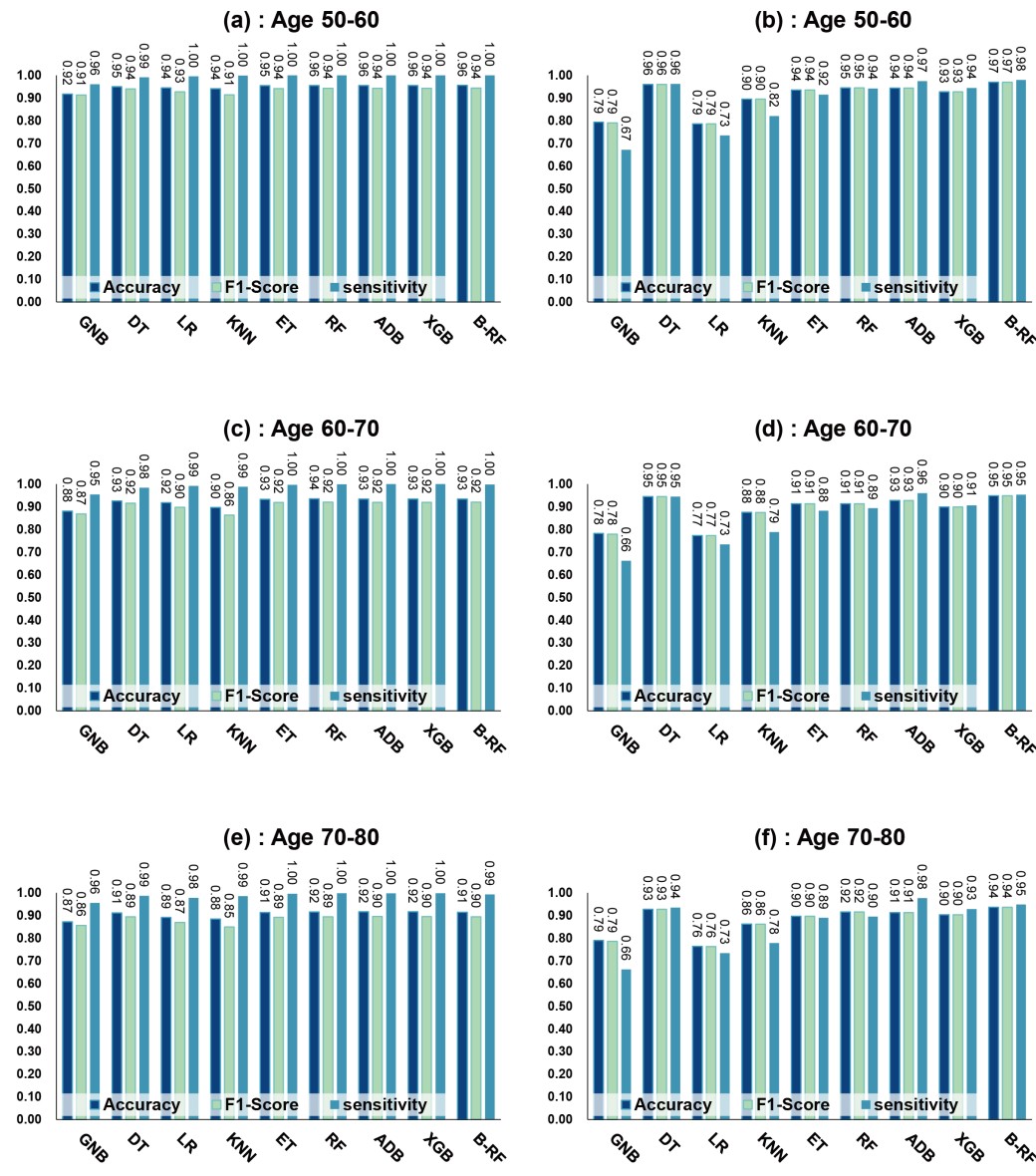

**Figure 7  Experimental chart of Diabetes Simple Diagnosis dataset.** (A) Age 50–60 groups without balanced data sets. (B) Age 50–60 groups balanced data sets. (C) Age 60–70 groups without balanced data sets. (D) Age 60–70 groups balanced data sets. (E) Age 70–80 groups without balanced data sets. (F) Age 70–80 groups balanced data sets. Note: Bagging-RF is abbreviated to B-RF.

## Summary of experimental results

According to the above results, the data-balanced Bagging-RF integrated learning classifier demonstrates superior performance in diabetes prediction on the Diabetes Simple Diagnosis and Diabetes Prediction Dataset for all three age groups: 50–60 years, 60–70 years, and 70–80 years. This not only validates the effectiveness of Bagging-DT-RF in handling unbalanced datasets but also highlights its robustness and superiority in predicting different age groups.

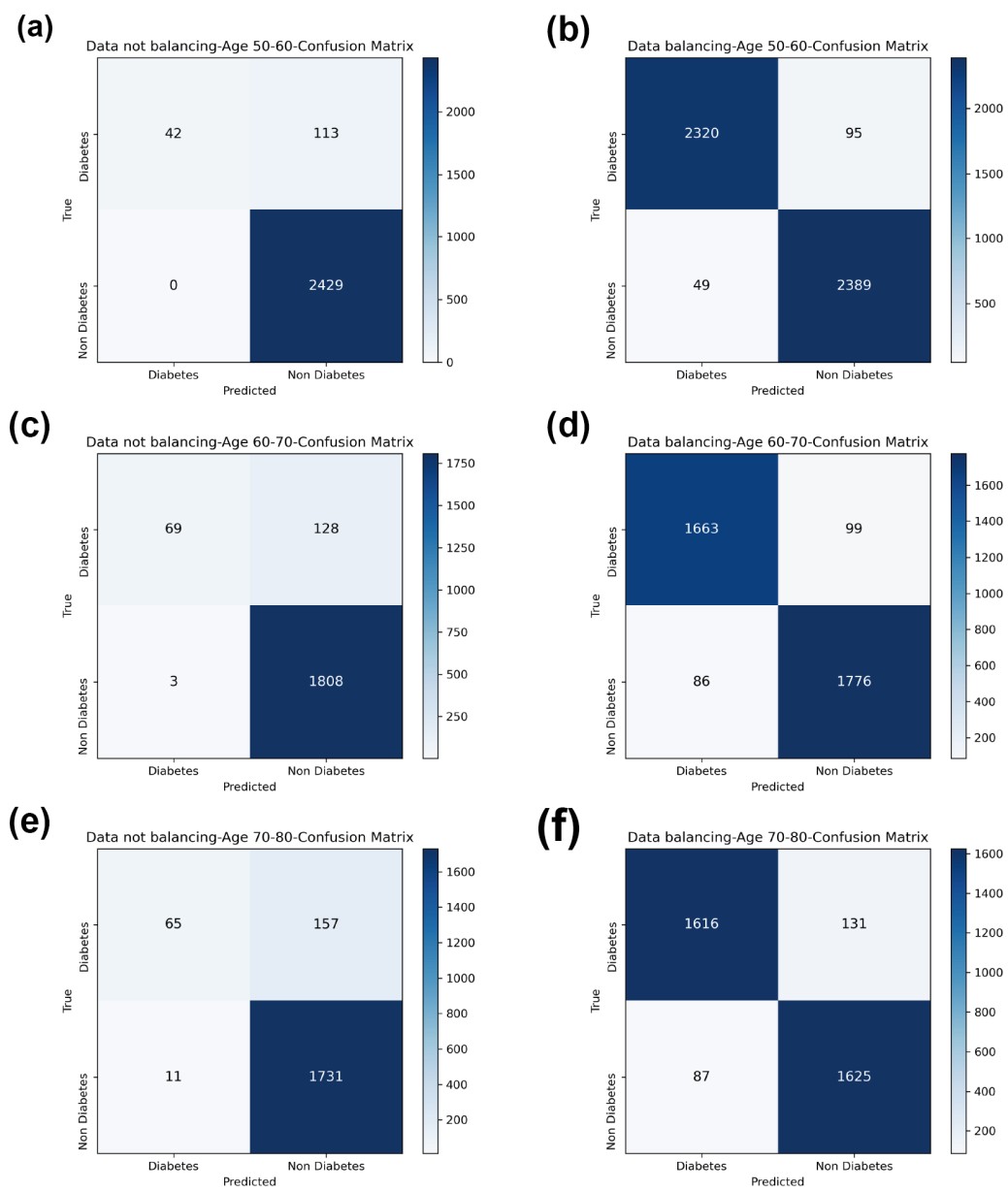

**Figure 8   Diabetes Simple Diagnosis confusion matrix.** (A) Age 50–60 groups without balanced confusion matrix. (B) Age 50–60 groups balanced confusion matrix. (C) Age 60–70 groups without confusion matrix. (D) Age 60–70 groups balanced confusion matrix. (E) Age 70–80 groups without confusion matrix. (F) Age 70–80 groups balanced confusion matrix.

After balancing with the SMOTE technique, the AUC values of each model increased to different degrees, indicating that the SMOTE technique effectively increased the representation of a few classes, thus improving the model's performance in recognising samples from other classes. The increase in AUC usually implies that the overall classification ability of the model is enhanced, especially in distinguishing samples from positive and negative classes. The percentage of non-diseased samples in the total population

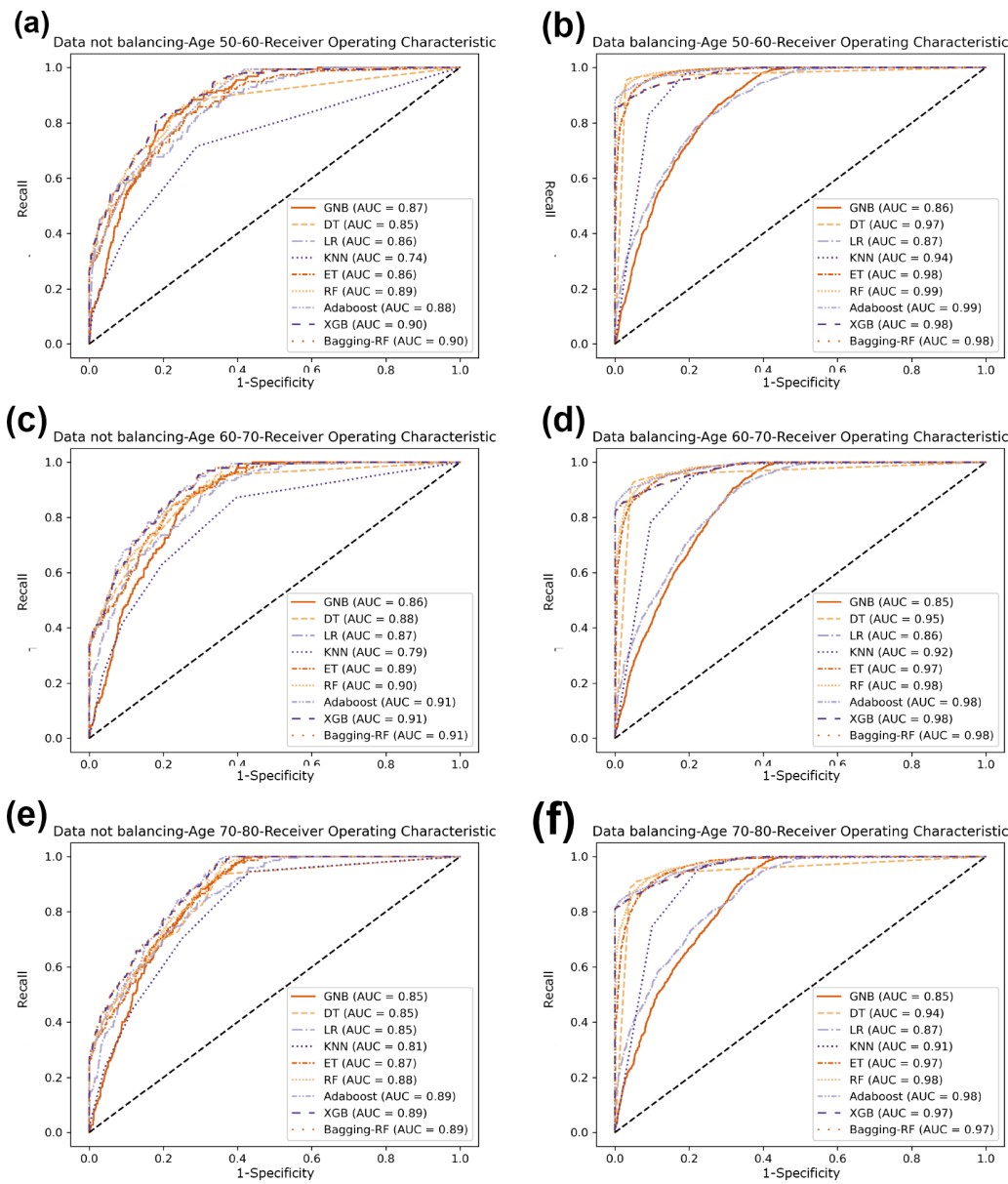

**Figure 9** **Diabetes Simple Diagnosis ROC curve.** (A) Age 50–60 groups without a balanced ROC curve. (B) Age 50–60 groups balanced ROC curve. (C) Age 60–70 groups without a balanced ROC curve. (D) Age 60–70 groups balanced ROC curve. (E) Age 70–80 groups without a balanced ROC curve. (F) Age 70–80 groups balanced ROC curve.

reduced following the data balancing process, which may be the reason why the specificity values of the models decreased to varied degrees following treatment with the SMOTE technique.

Except for the Bagging-RF model, the integrated learning models ET, RF, Adaboost, and XGB performed well, demonstrating the significant advantages of the integrated learning techniques in predicting diabetes.

## DISCUSSION

### Exploration of experimental results

An analysis of the performance of the machine learning classifiers in the two datasets for the three age groups found that the classifiers generally outperformed in predicting people in the 50–60 age group than they did for the 60–70 and 70–80 age groups. This suggests that the data in the 50–60 age group may have less noise or that the data in that age group is more abundant or of higher quality, making it easier for the classifiers to learn distinguishing features.

The performance of the Bagging-RF integrated learning model and the DT model was significantly improved after the data balancing process. This suggests that these two models can benefit from data balancing because they are better adapted to the adjusted data distribution, improving classification accuracy. Thus, the impact of data balancing treatments on classifier performance is complex and varied and needs to be carefully analysed and adapted to specific situations.

All models' specificity values declined, albeit to different degrees, following the data balancing procedure. This is because the balancing process grew proportionately to the dataset's original lower number of diseased samples and decreased proportionately to the initially higher number of non-diseased samples. This change in ratio from 1:10 to 1:1 changes the distribution of categories in the dataset, affecting the model's ability to recognise the non-diseased category and decreasing its specificity. Although data balancing may improve the model's identification of a few categories, it may also negatively affect the model's ability to identify the original majority of categories.

### Comparison with existing studies

The comparison of the proposed methodology with existing studies is detailed in Table 9. The table compares our study with previous studies in terms of accuracy, sample size, and research focus. Most of the earlier studies relied on the PIMA dataset, which contains only female samples, and the UCI-ERD dataset. Although the optimal classifiers on these datasets had accuracies of up to 99%, their studies focused mainly on the female population, and the sample sizes of the middle-aged and elderly populations were generally small. Fewer studies have been conducted in the field of diabetes detection for middle-aged and older adults, with only three related papers collected, and the datasets used in each of these studies vary. Comparing the accuracy of the models in these studies, the model proposed in this study demonstrated superior performance in predicting diabetes in middle-aged and older adults compared to the previous three studies.

## CONCLUSIONS

Early detection of diabetes mellitus in middle-aged and older adults can help with timely intervention, reduce the risk of complications, improve health and quality of life, and, at the same time, reduce the burden of medical care on individuals and society.

The Bagging-RF model proposed in this study was used to predict diabetes on the Diabetes Prediction Dataset and Diabetes Simple Diagnosis dataset for the middle-aged

**Table 9  Comparison of the proposed methodology with existing diabetes studies.**

| Method | Accuracy | Sample size | Research focus | Dataset |
|---|---|---|---|---|
| Logistic Regression (*Shaukat et al., 2023*) | 81.0% | 768 | Females | PIMA |
| DNN+ autoencoder (*Kannadasan, Edla & Kuppili, 2019*) | 86.3% | 768 | Females | PIMA |
| Fuzzy-KNN (*Haritha, Babu & Sammulal, 2018*) | 80.3% | 768 | Females | PIMA |
| ANN (*Waqas Khan et al., 2024*) | 99.3% | 520 | Everyone | PIMA |
|  | 98.9% | 768 | Females | UCI- ERD |
| Neural network (*Ma, 2020*) | 96.2% | 520 | Everyone | UCI-ERD |
| SVM + ANN (*Ahmed et al., 2022*) | 94.9% | 520 | Everyone | UCI-ERD |
| XGBoost (*Liu et al., 2022*) | 75.0% | 127,031 | Older people | 2019–2020 follow-up dataset |
| RF (*Wu et al., 2022*) | 92.0% | 17,833 | Older people | NHANES |
| Proposed model | 93.8%–97.5% | 36,843 | Middle-aged and Older people | Diabetes Prediction Dataset |
|  |  | 32,778 |  | Diabetes Simple Diagnosis |

and elderly age groups of 50–60, 60–70, and 70–80 years old, and the results showed excellent performance of the model. The results of this study will not only help individuals receive the necessary medical interventions in a timely manner to reduce the incidence of diabetes mellitus and its complications, but will also have a significant impact on the health and quality of life of the entire middle-aged and elderly population. While fully recognising the positive significance of this study, there are also limitations. First, there is still room for improvement in the performance of the proposed model. Although the model performs well in the prediction of 50–60-year-olds, there is still room for improvement in the accuracy of diabetes prediction for 60–70-year-olds and 70–80-year-olds. Second, the proposed model could be more interpretable, making it difficult for healthcare professionals and patients to understand its decision logic and prediction results. Third, the use of the SMOTE technique may introduce noise that negatively affects the model's performance.

In future research, the accuracy of machine learning models for diabetes prediction can be improved by investigating the following three aspects: First, feature engineering techniques such as PCA, SFS, or genetic algorithms can be applied to select critical features that significantly improve model performance. In this way, the model's feature set can be optimised, which, in turn, substantially improves the model's prediction accuracy. Second, hyper-parametric tuning techniques such as Bayesian optimisation and metaheuristics can be used to optimise the model parameters, thereby significantly improving the accuracy and reliability of the model in diabetes prediction tasks. Third, the SHAP model can be integrated into the diabetes prediction model to enhance its interpretability. The SHAP model can quantify each feature's contribution to the model's prediction results, which provides strong support for further optimisation of the model and the clinical decision-making of doctors.

**Abbreviations**

| KNN | K-Nearest Neighbours |
|---|---|
| LR | Logistic Regression |

**RF**     Random Forest
**ET**     Extra Trees
**DT**     Decision Tree
**Adaboost**  Adaptive Boosting
**XGB**    XGBoost
**GNB**    Gaussian Naive Bayes
**SMOTE**   Synthetic Minority Over-sampling Technique
**TP**     true positives
**TN**     true negatives
**FP**     false positives
**FN**     false negatives
**ROC**    Receiver Operating Characteristic
**AUC**    Area Under the Curve

## ACKNOWLEDGEMENTS

I would like to thank the providers of the diabetes dataset on the Kaggle website, as well as my family, friends, and the editors and reviewers of the article for their support and help with this research.

### Funding
The authors received no funding for this work.

### Competing Interests
The authors declare there are no competing interests.

### Author Contributions
- Yuanwu Shi performed the experiments, analyzed the data, prepared figures and/or tables, authored or reviewed drafts of the article, and approved the final draft.
- Jiuye Sun conceived and designed the experiments, performed the experiments, analyzed the data, performed the computation work, prepared figures and/or tables, authored or reviewed drafts of the article, and approved the final draft.

### Data Availability
The Diabetes Prediction Dataset is available at: https://www.kaggle.com/datasets/iammustafatz/diabetes-prediction-dataset.

The Diabetes Simple Diagnosis dataset is available at: https://www.kaggle.com/datasets/simaanjali/diabetes-simple-diagnosis.

The raw data are available in the Supplemental file.

### Supplemental Information
Supplemental information for this article can be found online at http://dx.doi.org/10.7717/peerj-cs.2436#supplemental-information.

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
