# Peer review of "Integrated bagging-RF learning model for diabetes diagnosis in middle-aged and elderly population"

_PeerJ Computer Science, doi:10.7717/peerj-cs.2436_

## Round 0.1 · original submission · Major Revisions

· Academic Editor

Major Revisions

Dear authors,

Thank you for submitting your article. Reviewers have now commented on your article and suggest major revisions. We do encourage you to address the concerns and criticisms of the reviewers and resubmit your article once you have updated it accordingly.

1. In general, the literature review is not sufficient. More recent literature should be explored in depth. It is more of the type “researcher X did Y” rather than an authoritative synthesis assessing the current state-of-the-art. Advantages and disadvantages of the related works should be evaluated. More recent literature should be deeply analyzed.
2. All of the values for the parameters of all algorithms selected for comparison should be given.
3. Bland character should be correctly used.
4. Limitations of the study should be discussed. Pros and cons of the methods should be clarified. What are the limitation(s) methodology(ies) adopted in this work? Please indicate practical advantages, and discuss research limitations.

Best wishes,

·

Basic reporting

no comment

Experimental design

This study could be significantly improved by answering the following questions. I have questions that hope the authors to respond:

1) The Introduction section should be improved by giving more information about the types of Diabetes mellitus (DM). Especially, please talk about the types of diabetes and ensemble learning.

2) The literature part of the study was shortly given in Introduction. However, this section could be improved by adding more ensemble learning algorithms to reveal the efficiency of ensemble learning algorithms.

3) Why authors did not employ stacked or super ensemble learning algorithm. Is there any reason?

4) The explanation of the employed machine learning algorithms is not detail. Please provide detailed information about the working principle of them.

5) Please provide a table that presents the hyperparameter settings of the employed machine learning algorithms in diagnosis of diabetic patients.

6) Please provide information about the robustness of the Bagging-DT-RF model.

7) What are the benefits and drawbacks of the study? Please present them in this study in a new sub-section.

Validity of the findings

no comment

Additional comments

Overall, this study purposes to develop a machine learning models to diagnosis diabetic patients. The highest accuracy score is obtained through the employed Bagging-DT-RF compared to the other models. This study has a contribution which merits attention and will be an interesting read for the journal readership. However, the study could be significantly improved by answering the questions that I wrote in my report.

Reviewer 2 ·

Basic reporting

A good number of typos are present; they need proper proofreading.

Experimental design

The authors should justify why they combined Bagging-DT-RF. How and why are the base learners (DTs) ensembled with Bagging and then applied RF? Needs clarification in this regard.

Validity of the findings

The authors need to justify the novelty of the work. Why are these classifiers employed? How is it superior to the previous research in the field they have considered with comparative analysis?

Additional comments

The authors aimed to concentrate on predicting diabetes within the 50-80 age bracket and introduce the Bagging-DT-RF ensemble learning model. This model exhibits exceptional predictive accuracy for diabetes in individuals aged 50-60, 60-70, and 70-80. Two diabetes datasets, Diabetes Simple Diagnosis, and Diabetes Prediction Dataset, available on Kaggle, were selected for analysis. The data were preprocessed, including solo heat coding, outlier removal, and age screening. The datasets were then categorized into age groups: 50-60, 60-70, and 70-80 years. Due to the uneven distribution of data after preprocessing, the dataset was balanced using the SMOTE technique. Subsequently, it was trained using an integrated model of Bagging-DT-RF and six basic machine learning classifiers: RF, GNB, LR, KNN, DT, and ET. The performance of each classifier was compared using four metrics: Accuracy, Precision, Recall, and F1 Score.
The comments are listed as follows:
1. The title of the research study is not attractive to the readers.
2. The authors have not added their affiliation in the manuscript.
3. The abstract should be concise, with a maximum of 250 words as follows: Background/Overview (2 or 3 Sentences), Motivation (2 or 3 Sentences), Objective (2 or 3 Sentences), Results in numerals (2 or 3 Sentences), Conclusion (2 Sentence), with only exactly one paragraph with no subheadings.
4. The literature survey subsection 1.2 should be specific to diabetes disease.
5. The caption for subsection 1.3 should be “objective.”
6. Motivation is not clear in the Introduction Section; it needs to be specified in a paragraph.
7. Figure 1 seems incomplete with its flow directions.
8. Why have the authors considered two datasets of similar diseases?
9. The authors should briefly provide the pre-processed dataset details in subsection 2.2 to show the effectiveness of the step.
10. The authors should briefly provide the pre-processed balanced dataset details in subsection 2.3 to show the effectiveness of the step.
11. Figures 2, 4, and 5 should be presented in a professional way.
12. The authors have considered very old references; they should consider recent papers.
13. The authors should justify why they combined Bagging-DT-RF. How and why are the base learners (DTs) ensembled with Bagging and then applied RF? Needs clarification in this regard.
14. Authors should consider Specificity as another performance measure and check whether it obtains a satisfactory outcome or not.
15. The figures should be included in good resolutions, properly captioned, and briefly described earlier in their presence with proper callouts.
16. The tables should be properly formatted, captioned, and briefly described earlier of their presence with proper callouts.
17. The equations should be properly formatted and briefly described earlier in their presence with proper callouts.
18. A good number of typos are present; they need proper proofreading.
19. The results and discussion section needs to be presented in a detailed way so that the reader can understand the obtained outcomes.
20. The authors need to justify the novelty of the work. Why are these classifiers employed?
21. How is it superior to the previous research in the field they have considered with comparative analysis?
22. The authors should provide two or three solid future directions of the study in the conclusion section using a paragraph.
23. The authors should change the Acknowledgements paragraph; they should need to thank those who have supported and encouraged in this research.

---

## Round 0.2 · Minor Revisions

· Academic Editor

Minor Revisions

Dear authors,

Thank you for your paper. According to one reviewer, your paper still needs revision and we encourage you to address the concerns and criticisms of Reviewer 2 and resubmit your article once you have updated it accordingly.

Best wishes,

·

Basic reporting

The authors provided sufficient responses for my concerns.

Experimental design

I am happy about the experimental design and the results.

Validity of the findings

The obtained statistical scores show the robustness of the prosed ensemble model.

Additional comments

The manuscript has been significantly improved with the current version.

Reviewer 2 ·

Basic reporting

1. Change “scholars” to “researchers” or “authors”.
2. Change the title to “Integrated Bagging-RF Learning Model for Diabetes Diagnosis in Middle-aged and Elderly Population”
3. In sub-section 1.3, change to “Objective” in place of “objective”.
4. There is no need to give a sub-section for Research Background; you can start with the paragraph, whereas the sub-section should start with “1.1. Motivation” and so on.
5. Recall, Sensitivity, and true positive rate (TPR) all the same. Authors need to consider anyone in this study.
6. Specificity and true negative rate (TNR), both are same. Authors need to consider anyone in this study.
7. According to points 5 and 6, the authors need to change the contents of the manuscript along with the obtained result tables and figures.

Experimental design

Recall, Sensitivity, and true positive rate (TPR) all the same. Authors need to consider anyone in this study.
Specificity and true negative rate (TNR), both are same. Authors need to consider anyone in this study.
According to points 5 and 6, the authors need to change the contents of the manuscript along with the obtained result tables and figures.

Validity of the findings

Recall, Sensitivity, and true positive rate (TPR) all the same. Authors need to consider anyone in this study.
Specificity and true negative rate (TNR), both are same. Authors need to consider anyone in this study.
According to points 5 and 6, the authors need to change the contents of the manuscript along with the obtained result tables and figures.

Additional comments

No More Comments

---

## Round 0.3 · accepted · Accept

· Academic Editor

Accept

Dear authors,

Thank you for clearly addressing all the concerns and criticisms of the reviewers. Your paper seems to be improved and is now acceptable for publication in light of this last revision.

Best wishes,

Reviewer 2 ·

Basic reporting

There is no need to give a sub-section heading to the first paragraph under the Introduction Section.
The second paragraph can given with the sub-section heading as “1.1. Motivation”.

Experimental design

No more comments

Validity of the findings

No more comments

Additional comments

No more comments